# TOWARDS UNIFIED AND EFFECTIVE DOMAIN GENERALIZATION

## ABSTRACT

We propose **UniDG**, a novel and **Uni**fied framework for **D**omain **G**eneralization that is capable of significantly enhancing the out-of-distribution generalization performance of foundation models regardless of their architectures. The core idea of UniDG is to finetune models during inference stage which saves the cost of iterative training. Specifically, we encourage models to learn the distribution of testing data in an unsupervised manner and impose a penalty regarding the updating step of model parameters. The penalty term can effectively reduce the catastrophic forgetting issue as we would like to maximally preserve the valuable knowledge in the original model. Empirically, across 12 visual backbones, including CNN-, MLP-, and transformer-based models, ranging from 1.89M to 303M parameters, UniDG shows an average accuracy improvement of +5.4% on DomainBed. We believe that these performance results are able to manifest the superiority and versatility of UniDG.

## 1 INTRODUCTION

The Out-Of-Distribution (OOD) problem is a prevalent topic in the machine learning and computer vision communities (Long et al., 2015; Saito et al., 2020; Sun & Saenko, 2016; Ebrahimi et al., 2020) as models of various architectures and scales are suffering from this problem (Zhou et al., 2022; Li et al., 2023; Chen et al., 2022a; Peng et al., 2022). Therefore, training deep models to generalize well on new domains has become a prevalent research topic (Long et al., 2015; Li et al., 2018b; Wang et al., 2019; Chen et al., 2022b; Cha et al., 2021; 2022).

To overcome the domain shift problem, pretraining-based methods (Radford et al., 2021; Singh et al., 2022; Cha et al., 2022) utilize large-scale data to obtain better generalization ability. However, in practice, domain shift can be so significant that even though the powerful foundation models have been pretrained on huge-scale datasets, directly generalizing the models to new domains still delivers unsatisfactory performance, as shown in Figure 1. Another drawback of pretraining-based methods is the inferior finetuning performance - finetuning pretrained models leads to catastrophic forgetting and limited improvement on new domains (Cha et al., 2022; Li et al., 2022; Chen et al., 2022c). As a workaround, pretraining-based methods may add data from the new domains into the pretraining dataset and retrain the models from scratch (Shu et al., 2023). When the pretraining dataset is large (*e.g.*, CLIP (Radford et al., 2021) uses LAION-400M), this approach becomes significantly expensive.

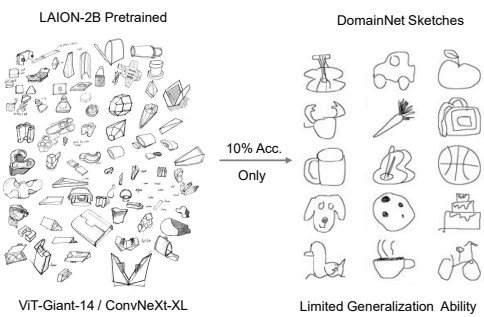

Figure 1: Large-scale pretrained foundation models are still suffering from domain shifts.

In contrast to the pretraining-based methods, Test-Time Adaptation (*TTA*) (Sun et al., 2020; Wang et al., 2021; 2022b;a) is an alternative to mitigate domain shift on new domains. First, TTA requires no pretraining with novel data, and can directly leverage the off-the-shelf models. Second, by updating parameters in both training and evaluation stages (Sun et al., 2020), TTA reduces the reliance of models on annotations in new domains. However, we would like to note several drawbacks of

existing TTA methods. **1)** Most TTA methods (Wang et al., 2021; Iwasawa & Matsuo, 2021; Jang & Chung, 2023) require updating Batch Normalization (BN) (Ioffe & Szegedy, 2015) layers in the original model to adapt to the distribution of test data. However, recent visual foundation models such as vision transformers (Dosovitskiy et al., 2020) are developed with Layer Normalization (LN) layers. Due to the essential difference between BN and LN, simply adapting the ideas of BN-based methods to LN layers results in minimal improvement (around 0.5%, see Appendix § F). **2)** Recent TTA methods (Zhang et al., 2023b; Park et al., 2023; Zhang et al., 2023a; Chen et al., 2023) show limited scalability on common visual foundation models ranging from small to large scales.

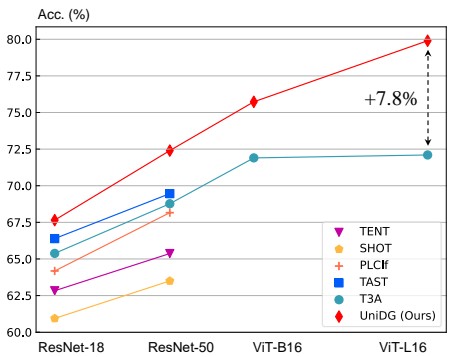

Figure 2: A comparison between existing methods and UniDG on the accuracy averaged across the PACS, VLCS, Office-Home, and TerraInc datasets.

For example, only limited improvements (less than 2%) on large-scale foundation models (Radford et al., 2021; Liu et al., 2022) are observed. **3)** From a theoretical perspective, we find these TTA methods reduce the Neural Tangent Kernel (Jacot et al., 2018) in the adaptation process, which limits the further generalization (theoretical analysis is presented in the Appendix § B).

To address the aforementioned drawbacks, we focus on an important topic of TTA method - the appropriate way to update the encoder (*i.e.*, feature extractor) for TTA. Prior works either update the encoder via backpropagation or freeze it, but either way has its weaknesses. 1) If we allow the encoder to update, similar to the weakness of finetuning a pretrained encoder, which is discussed above, catastrophic forgetting can happen during TTA and result in a significantly lower quality of extracted features. 2) With the encoder frozen, the encoder cannot well adapt to the new domains hence the extracted features have to be refined with extra mechanisms, in order to be well utilized by the classifier.

In this paper, we propose a novel method, named **Marginal Generalization**, to update the encoder for TTA. Intuitively, Marginal Generalization aims to let the encoder learn representations of the target data *within a certain distance from the representations obtained by the initial model*. Here we use a simplified notation for brevity. Let $\sigma$ be the specified distance, $f(\cdot)$ be the fixed initial encoder and $f'(\cdot)$ be a learnable copy of $f(\cdot)$, $x$ be the samples of the target domain, $q(\cdot)$ be the classifier which takes the representations $f'(x)$ as inputs, the objective is to

$$\text{minimize} \quad [\text{entropy}(\text{softmax}(q(f'(x))))] \quad s.t. \ \|f'(x) - f(x)\|_F \leq \sigma \,. \tag{1}$$

By doing so, we overcome the drawbacks of the aforementioned two traditional approaches.

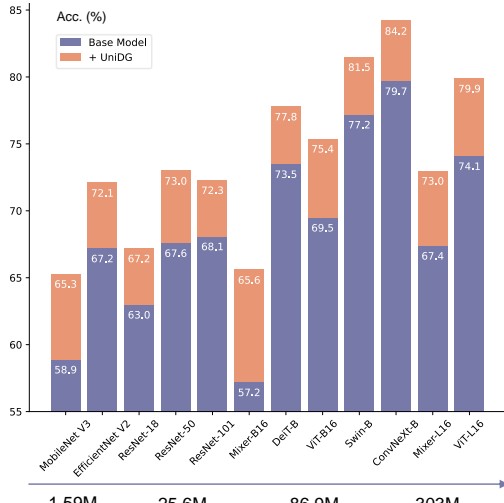

Figure 3: UniDG brings out an average of 5.4% improvement to 12 backbones which scale from 1.59M to 303M parameters.

1) Intuitively, while the encoder $f'(\cdot)$ is trying to adapt to the novel data, it always refers to the original model $f(\cdot)$ and keeps the representations within a distance $\sigma$ from the original, which means the pretrained source knowledge can be preserved and catastrophic forgetting is avoided. 2) As we keep updating the encoder via entropy minimization on the test data, it cooperates better with the classifier and yields more discriminative features on the target domain.

We would like to note that Marginal Generalization is **uni**versal because it does not require any specific structures in the original model nor the properties of the data, as well as effective (achieving improvement of 3.3% on average accuracy as shown in Table 4). In addition, the features extracted by the updated encoder can be utilized by multiple TTA mechanisms. For example, by naturally combining Marginal Generalization and Memory Bank (Wu et al., 2018), we propose **Differentiable Memory Bank**, which demonstrates

superior performance over the traditional memory bank methods because it performs feature filtration and storage on differentiable features. For example, compared with T3A (Iwasawa & Matsuo, 2021) that adopts typical memory bank, our method with ResNet-50 backbone leads it by 4.3% on average accuracy across 4 datasets shown in Table 2. Intuitively, UniDG simultaneously utilizes the local minimum of distances between adapted and source representations and the local maximum of information entropy between adapted representations and pseudo labels in the test-time process to continuously approximate the models on the test data with reserved pretrained knowledge. The details will be presented in Section 2.2, and 2.3.

Based on Marginal Generalization, we propose a framework composed of an adaptation method of the encoder (which is a **uni**versal method to extract better features) and Differentiable Memory Bank (which is a **uni**versal mechanism to refine features for DG) so that the framework is named **UniDG**, which delivers state-of-the-art performance on multiple domain generalization benchmarks. For example, UniDG delivers an average accuracy of 79.6% on 5 widely adopted benchmarks including PACS, VLCS, and OfficeHome, outperforming the second-best CAR-FT (Mao et al., 2022) by 1.0%. Additionally, UniDG is an architecture-agnostic framework that consistently yields significant improvements when applied to a wide range of visual backbones, including models of varying scales such as MobileNet V3 (Howard et al., 2019), ConvNeXt-Base (Liu et al., 2022), and ViT-Large (Dosovitskiy et al., 2020), demonstrating its strong scalability. For example, UniDG improves the mean accuracy scores by 5.4% with such 12 models on PACS (Torralba & Efros, 2011), VLCS (Li et al., 2017), OfficeHome (Venkateswara et al., 2017), and TerraInc (Beery et al., 2018). We would like to note that Marginal Generalization and Differentiable Memory Bank can also be used separately and combined with other methods. When we combine these two schemes, we observe an average improvement of +5.0%.

Our contributions are summarized as follows.

- We propose Marginal Generalization, which addresses the problem of adapting the encoder for TTA.

- With Marginal Generalization, we naturally upgrade the traditional memory bank mechanism to Differentiable Memory Bank and propose a universal TTA framework named UniDG.

- UniDG consistently outperforms the previous state-of-the-art methods by a significant margin (+5.4% on DomainBed). It applies to a wide range of models of different architectures and varying scales, consistently resulting in satisfactory TTA performance.

- We show that UniDG's components can also be separately combined with other methods, demonstrating its flexibility.

## 2 METHOD

We first introduce the formulation of domain generalization and test-time adaptation in § 2.1. The framework of UniDG comprises two components: 1) we employ *Marginal Representation Generalization* (§ 2.2) to adapt the encoder, 2) we utilize prototypes with *Differentiable Memory Bank* (§ 2.3) for learning a discriminative classifier on the target domain.

### 2.1 PRELIMINARY

**Domain Generalization.**  Given a set of source domains $\mathcal{D}_S = \{\mathcal{D}_1, \mathcal{D}_2, \cdots, \mathcal{D}_N\}$, each domain $\mathcal{D}_j$ contains images and labels, $\{(\boldsymbol{x}_i, y_i)\}_{i=1}^{\|\mathcal{D}_j\|}$, where $\boldsymbol{x}_i$ denotes an image and $y_i$ indicates the corresponding ground truth label, the goal of DG is to generalize models on a novel target domain $\mathcal{D}_T$ that is different from any of the source domains by training on $\mathcal{D}_S$. We denote the mapping function of the model as $\mathcal{F} : \boldsymbol{x} \to \boldsymbol{p} \in \mathbb{R}^C$, where $\boldsymbol{p}$ is the prediction and $C$ is the number of categories. $\mathcal{F}$ comprises two steps: feature extraction with the encoder $f(\cdot)$ and prediction with the classifier $q(\cdot)$ based on the features. Let $\theta$ be the parameters, $\mathcal{F}$ can be formulated as $\mathcal{F}(\boldsymbol{x}; \theta) = q(f(\boldsymbol{x}))$.

**Training on source domains.** We use $\ell_{\text{CE}}(\cdot)$ to denote the cross-entropy function, and the objective of training on the source domains is to optimize $\theta$ as

$$\theta^* = \arg\min_\theta \mathbb{E}_{(\boldsymbol{x}, y) \in \mathcal{D}_S} [\ell_{\text{CE}}(\mathcal{F}(\boldsymbol{x}; \theta), y)]. \tag{2}$$

**Test-Time Adaptation.** With $\theta^*$ trained on the source domains $\mathcal{D}_S$, test-time adaptation is a self-supervised learning process to further adapt parameters to the target domain $\mathcal{D}_T$. The encoder parameters during test time can be optimized as the following, where $\ell_{\text{TTA}}(\cdot)$ is the softmax entropy:

$$\theta' = \arg\min_\theta \mathbb{E}_{(\boldsymbol{x}) \in \mathcal{D}_T} [\ell_{\text{TTA}}(\mathcal{F}'(\boldsymbol{x}; \theta))]. \tag{3}$$

## 2.2 MARGINAL GENERALIZATION

Marginal Generalization aims to constrain the discrepancy between features extracted by the source encoder $f$ and the adapted encoder $f'$ during the adaptation process so that the adapted model will be able to maintain general representation bias and relieve catastrophic forgetting while updating parameters. Here we adopt Euclidean distance as the metric out of its simplicity and universality, which is formulated with the Frobenius norm $\| \cdot \|_F$. We use $\theta_e$ to denote the parameters of the encoder, which is a subset of $\theta$, so that the encoder can be formulated as $f(\cdot; \theta_e)$. Given the pre-defined distance threshold $\sigma$, the objective then becomes

$$\theta' = \arg\min_\theta \mathbb{E}_{(\boldsymbol{x}) \in \mathcal{D}_T} [\ell_{\text{TTA}}(\mathcal{F}'(\boldsymbol{x}; \theta))] \quad s.t. \ \|f'(x; \theta_e') - f(x; \theta_e)\|_F \leq \sigma. \tag{4}$$

The motivation is that we desire to gradually update the parameters of the adapted encoder under the condition that the representation bias will not get sharply adapted. For the source feature extractor $f(\cdot; \theta_e)$, we freeze it and still use it to extract the representation from target domains as pretrained knowledge. For the adapted encoder $f'(\cdot; \theta_e')$, we initialize it with the source-pretrained parameters $\theta_e$. Therefore, the discrepancy between the original and adapted representations can be formulated as the distance between $f(x; \theta_e)$ and $f'(x; \theta_e')$.

To approximate such a hard constraint with a back-propagation-based method, we propose a novel loss function named *Marginal Adaptation Loss* to constrain the update of the parameters of the encoder. The Marginal Adaptation Loss can be formulated as:

$$\mathcal{L}_m = \frac{1}{\|\mathcal{D}_T\|} \sum_{i=1}^{\|\mathcal{D}_T\|} \max(\|f'(\boldsymbol{x}_i; \theta_e') - f(\boldsymbol{x}_i; \theta_e)\|_F^2 - \sigma, 0). \tag{5}$$

The update of parameters of the classifier $q(\cdot)$ and encoder is guided by the entropy on the target domain. Based on the extracted representations $f'(\boldsymbol{x}; \theta_e')$, we use a linear layer to work as a classifier and obtain the classification probability $\boldsymbol{p} = \texttt{softmax}(q'(f'(\boldsymbol{x}, \theta_e')))$ using a $\texttt{softmax}$ operation. Then we take the entropy as the loss function to derive the gradients for updating the classifier and encoder, through which we can introduce the probabilistic distribution of target domains to our classifier:

$$\mathcal{L}_e = -\frac{1}{N_b} \sum_{i=1}^{N_b} \sum_{c=1}^{C} \boldsymbol{p}_c \log \boldsymbol{p}_c. \tag{6}$$

## 2.3 DIFFERENTIABLE MEMORY BANK

With Marginal Generalization, we are able to learn a well-adapted encoder that can extract discriminative features on the target domain. However, since there is no labeled data on the target domain, only training with the unsupervised losses $\mathcal{L}_m$ and $\mathcal{L}_e$ is hard to get a classifier $q(\cdot)$ with high performance on the target. To mitigate this issue, we propose to update the classifier with a differentiable memory bank. We utilize the memory bank to select prototypes suitable for the new domain, develop class-wise prototypes directly differentiable with loss function, and update the whole bank in every forward step.

**Class-wise prototypes** are stored in the memory bank in order to enhance the classifier. Specifically, for each class $j$, the prototype $\boldsymbol{v}_j$ is initialized with the corresponding weights of the source

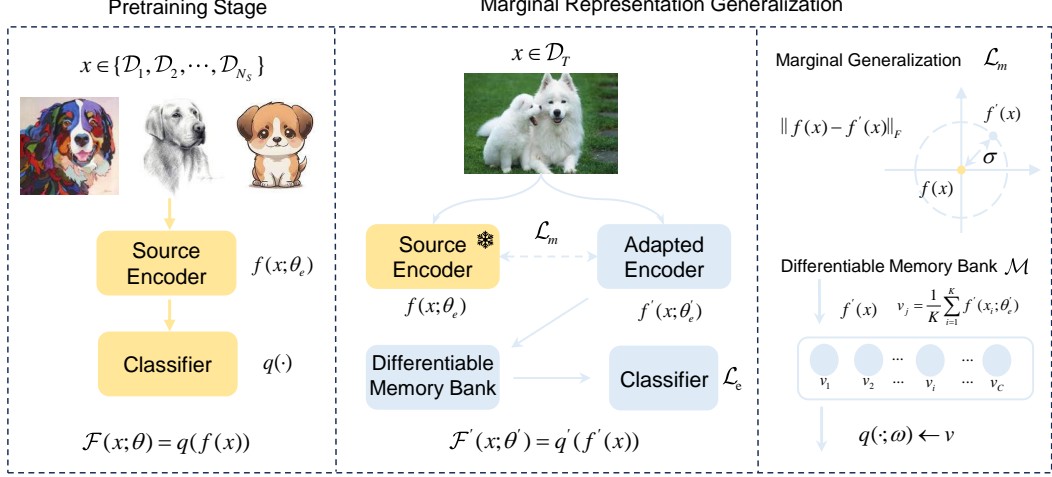

Figure 4: Illustration of UniDG, which consists of Marginal Generalization (§ 2.2) and the Differentiable Memory Bank mechanism (§ 2.3).

classifier layer. In the self-supervised adaptation process, for each target sample $\boldsymbol{x}$, we extract the representations $f'(\boldsymbol{x})$ and obtain the output of classifier $q'(f'(\boldsymbol{x}; \theta'); \omega)$. Then we predict pseudo labels $\hat{y} = \arg\max[\texttt{softmax}(q'(f'(\boldsymbol{x})))]$ and utilize the entropy between representations and pseudo labels as the criterion to select the Top-$K$ instances of each class with highest classification confidence, where $K$ is a pre-defined hyper-parameter. After that, we utilize the representations of the Top-$K$ samples to produce the class-wise prototypes $\boldsymbol{v}_j = \frac{1}{K} \sum_{i=1}^{K} f'(\boldsymbol{x_i})$.

**Memory bank** is set to store the prototypes of each class $\mathcal{M} = \bigcup_{j=1}^{C} \{\boldsymbol{v}_j\}, \boldsymbol{v}_j \in \mathbb{R}^d$, where $\mathcal{M}$, $C$, and $d$ denote the memory bank, the number of classes, and feature dimension. At each forward step, we compute the prototypes, which will be further used to update the classifier weights $\omega$. For a given sample $\boldsymbol{x}$ with feature $\boldsymbol{z} = f'(\boldsymbol{x}; \theta)$, the classification probability of class $j$ can be computed as:

$$\boldsymbol{p}_j = \frac{\exp(\boldsymbol{z} \cdot \omega^j)}{\sum_k \exp(\boldsymbol{z} \cdot \omega^k)}, \quad \omega^k \in \mathbb{R}^d \tag{7}$$

where $\omega^j$ is the $j$-th element of the weight matrix $\omega$. Note that for $q(\cdot; \omega)$ to classify target samples correctly, the weight $\omega^j$ needs to be representative of features of the corresponding class $j$. This indicates that the meaning of $\omega^j$ coincides with the ideal cluster prototype of class $j$ in the target domain. Thus, we propose to use the estimate of the ideal target cluster prototypes $\{\boldsymbol{v}_j\}_{j=1}^{C}$ to update the classifier weights: $\omega^j \leftarrow \boldsymbol{v}_j$. This process is essential in learning a robust classifier for the target domain with no labeled data.

## 2.4 UniDG Learning for Domain Generalization

In UniDG framework, the marginal generalization is proposed to learn a well-adapted feature encoder without catastrophic forgetting, and the differentiable memory bank is proposed to learn a discriminative classifier for the target domain. While updating $\omega$ with target prototypes, the overall learning objective is:

$$\mathcal{L}_{\text{UniDG}} = \mathcal{L}_e + \lambda \cdot \mathcal{L}_m \tag{8}$$

## 3 Experiments

### 3.1 Setup

**Dataset** VLCS (Torralba & Efros, 2011) contains 10,729 instances of 5 classes derived from four photographic datasets in accordance with different domains. PACS (Li et al., 2017) comprises four

Table 1: Overall out-of-domain accuracies with train-validation selection criterion on the **DomainBed** benchmark. The best result is highlighted in **bold**. UniDG achieves the best performances on PACS, VLCS, OfficeHome, TerraIncognita, and DomainNet datasets.

| Algorithm | Venue | Pretraining | PACS | VLCS | OfficeHome | TerraInc | DomainNet | Avg. |
|---|---|---|---|---|---|---|---|---|
| ERM (ResNet-50) (Vapnik, 1991) | | | 85.7 ± 0.5 | 77.4 ± 0.3 | 67.5 ± 0.5 | 47.2 ± 0.4 | 41.2 ± 0.2 | 63.8 |
| DANN (Ganin et al., 2016) | JMLR'16 | | 84.6 ± 1.1 | 78.7 ± 0.3 | 68.6 ± 0.4 | 46.4 ± 0.8 | 41.8 ± 0.2 | 64.0 |
| MMD (Li et al., 2018b) | CVPR'18 | | 85.0 ± 0.2 | 76.7 ± 0.9 | 67.7 ± 0.1 | 42.2 ± 1.4 | 39.4 ± 0.8 | 62.2 |
| IRM (Arjovsky et al., 2019) | ArXiv'20 | | 83.5 ± 0.8 | 78.5 ± 0.5 | 64.3 ± 2.2 | 47.6 ± 0.8 | 33.9 ± 2.8 | 61.6 |
| FISH (Shi et al., 2021) | ICLR'22 | | 85.5 ± 0.3 | 77.8 ± 0.3 | 68.6 ± 0.4 | 45.1 ± 1.3 | 42.7 ± 0.2 | 63.9 |
| SWAD (Cha et al., 2021) | NeurIPS'21 | | 88.1 ± 0.1 | 79.1 ± 0.1 | 70.6 ± 0.2 | 50.0 ± 0.3 | 46.5 ± 0.1 | 66.9 |
| ERM (ViT-S/16) (Dosovitskiy et al., 2020) | ICLR'21 | | 86.2 ± 0.1 | 79.7 ± 0.0 | 72.2 ± 0.4 | 42.0 ± 0.8 | 47.3 ± 0.2 | 65.5 |
| Fishr (Rame et al., 2022) | ICML'22 | ImageNet | 85.5 ± 0.2 | 77.8 ± 0.2 | 68.6 ± 0.2 | 47.4 ± 1.6 | 41.7 ± 0.0 | 64.2 |
| MIRO(Cha et al., 2022) | ECCV'22 | | 85.4 ± 0.4 | 79.0 ± 0.6 | 70.5 ± 0.4 | 50.4 ± 1.1 | 44.3 ± 0.2 | 65.9 |
| GMoE-S/16 (Li et al., 2023) | ICLR'23 | | 88.1 ± 0.1 | 80.2 ± 0.2 | **74.2 ± 0.4** | 48.5 ± 0.4 | 48.7 ± 0.2 | 67.9 |
| ITTA (Chen et al., 2023) | CVPR'23 | | 83.8 ± 0.3 | 76.9 ± 0.6 | 62.0 ± 0.2 | 43.2 ± 0.5 | 34.9 ± 0.1 | 60.2 |
| DomainAdaptor (Zhang et al., 2023a) | ICCV'23 | | 84.9 ± 0.2 | 78.50 ± 0.2 | 66.7 ± 0.3 | - | - | - |
| AdaNPC (Zhang et al., 2023b) | ICML'23 | | 88.9 ± 0.1 | 80.2 ± 0.2 | 66.3 ± 0.1 | **54.0 ± 0.1** | 43.1 ± 0.8 | 66.5 |
| UniDG | Ours | | **89.0 ± 0.3** | **81.6 ± 0.1** | 68.9 ± 0.1 | 52.9 ± 0.2 | **50.2 ± 0.1** | **68.5** |
| *UniDG with Existing DG Methods* | | | | | | | | |
| CORAL (Sun & Saenko, 2016) | ECCV'16 | ImageNet | 86.0 ± 0.2 | 77.7 ± 0.5 | 68.6 ± 0.4 | 46.4 ± 0.8 | 41.8 ± 0.2 | 64.1 |
| UniDG + CORAL | Ours | ImageNet | 89.2 ± 0.2 | 82.1 ± 0.1 | 70.6 ± 0.1 | 53.0 ± 0.2 | 51.3 ± 0.1 | 69.3 +5.2 |
| MIRO(Cha et al., 2022) | ECCV'22 | ImageNet | 85.4 ± 0.4 | 79.0 ± 0.6 | 70.5 ± 0.4 | 50.4 ± 1.1 | 44.3 ± 0.2 | 65.9 |
| UniDG + MIRO | Ours | ImageNet | 90.4 ± 0.4 | 84.1 ± 0.2 | 72.5 ± 0.4 | 54.4 ± 0.5 | 52.6 ± 0.2 | 70.8 +4.9 |
| *ViT-B/16 (Dosovitskiy et al., 2020) Backbone* | | | | | | | | |
| ERM (Vapnik, 1991) | | | 93.7 | 82.7 | 78.5 | 52.3 | 53.8 | 72.2 |
| MIRO (Cha et al., 2022) | ECCV'22 | | 95.6 | 82.2 | 82.5 | 54.3 | 54.0 | 73.7 |
| DPL (Zhang et al., 2021) | Arxiv'22 | CLIP | 97.3 | 84.3 | 84.2 | 52.6 | 56.7 | 75.0 |
| CAR-FT (Mao et al., 2022) | Arxiv'22 | | 96.8 | 85.5 | 85.7 | 61.9 | **62.5** | 78.5 |
| UniDG | Ours | | 96.7 ± 0.4 | 86.3 ± 0.2 | 86.2 ± 0.1 | 62.4 ± 0.2 | 61.3 ± 0.2 | 78.6 |
| *Base-scale Visual Backbone* | | | | | | | | |
| ERM (Vapnik, 1991) | | | 89.6 ± 0.4 | 78.6 ± 0.3 | 71.9 ± 0.6 | 51.4 ± 1.8 | 48.5 ± 0.6 | 68.0 |
| MIRO (Cha et al., 2022) (RegNetY-16GF (Radosavovic et al., 2020)) | ECCV'22 | SWAG | **97.4 ± 0.2** | 79.9 ± 0.6 | 80.4 ± 0.2 | 58.9 ± 1.3 | 53.8 ± 0.1 | 74.1 |
| UniDG + CORAL (Sun & Saenko, 2016) + ConvNeXt-B (Liu et al., 2022) | Ours | ImageNet | 95.6 ± 0.2 | **84.5 ± 0.1** | **88.9 ± 0.3** | **69.6 ± 0.7** | 59.5 ± 0.1 | **79.6** |

domains: art, cartoons, photos, and sketches, including 9,991 instances of classes. OfficeHome (Venkateswara et al., 2017) derives from domains like art, clipart, product, and real, containing 15,588 images of 65 classes. TerraIncognita (Beery et al., 2018)is a real-world dataset that collects photos of wild animals taken by cameras at different locations. It contains 24,788 photos of 10 classes according to the species of animals. DomainNet (Peng et al., 2019) is the largest dataset for domain generalization tasks, including 6 domains, 345 classes, and a total of 586,575 images.

**Evaluation Metric** We evaluate UniDG by taking 3 parallel trials with random seeds to calculate means and standard errors of classification accuracy (%) on 5 datasets. There are 22 different novel environments to evaluate the abilities of the network for generalization. We report detailed results for each environment in Appendix F.

**Implementation Details** All experimental results are conducted on NVIDIA A100 GPUs. If not specified, we utilize ResNet-50 (He et al., 2016) for extracting visual features and a single classifier for classification. On test-time benchmarks, we utilize ERM (Vapnik, 1991) algorithm as our default method for training source models. We also follow default hyper-parameters of DomainBed (Gulrajani & Lopez-Paz, 2020) like initial learning rate of $5e-5$, weight decay of $0.0$, batch size of $32$, holdout fraction of $0.2$, and $\sigma$ of $0.15$ (see Appendix § C for more discussions).

## 3.2 MAIN RESULTS

We report experimental results on the domain generalization (§ 3.2.1) and test-time adaptation benchmarks (§ 3.2.2). UniDG delivers new state-of-the-art performances on such benchmarks.

### 3.2.1 DOMAIN GENERALIZATION BENCHMARKS

UniDG prominently achieves a brilliant performance on Domain generalization tasks. Table 1 shows the performances of the existing advanced approaches for DG tasks using different pre-training methods. The upper part of the table demonstrates that with ImageNet pre-training, UniDG significantly outperforms various classic models and shows satisfactory stability. Specifically, it achieved an average accuracy of 68.5 on VLCS, PACS, OfficeHome, Terrain, and DomainNet, exceeding AdaNPC by +2.0%, and the best results on VLCS, PACS, terrain, and DomainNet. The remaining part of Table 1 shows more results with large-scale CLIP and SWAG pre-training. Expectedly, the CLIP- and SWAG-trained models outperform the traditional ImageNet-trained ones. However, impressively, with only ImageNet pre-training, UniDG outperforms the CAR-FT model with CLIP pre-training by 1.1% in the average accuracy (79.6% vs. 78.5%). On the terrain data set with complex domain shift, the accuracy of UniDG reached 62.4%, outperforming CAR-FT by 0.5%.

Table 2: Average accuracy (%) using classifiers learned by ERM on the domain generalization benchmarks. We use ResNet-18/50 as backbones. **Bold** indicates the best for each benchmark.

| Generalization Algorithm | Test-Time Algorithm | Backbone | VLCS | PACS | OfficeHome | TerraIncognita | Avg |
|---|---|---|---|---|---|---|---|
| CLIP (Radford et al., 2021) | Zero-Shot | ViT-B16 | 82.6±0.0 | 95.6±0.0 | 79.1±0.0 | 31.1±0.0 | 72.2 |
| ERM (Vapnik, 1991) | + None | ResNet-18 | 74.9±0.5 | 79.3±0.8 | 62.1±0.3 | 40.6±1.2 | 64.2 |
| | + PL [ICMLW'13] (Lee, 2013) | | 63.0±2.7 | 71.0±1.8 | 58.2±3.2 | 37.4±7.2 | 57.4 |
| | + PLClf [ICMLW'13] (Lee, 2013) | | 74.9±0.6 | 78.1±2.3 | 61.9±0.4 | 41.8±1.9 | 64.2 |
| | + SHOT [ICML'20] (Liang et al., 2020) | | 65.2±2.3 | 82.4±0.6 | 62.6±0.4 | 33.6±1.0 | 60.9 |
| | + Tent [ICLR'21] (Wang et al., 2021) | | 72.9±0.8 | 83.9±0.5 | 60.9±0.4 | 33.7±1.1 | 62.8 |
| | + TentBN [ICLR'21] (Wang et al., 2021) | | 67.0±1.2 | 80.8±1.0 | 62.6±0.4 | 40.0±0.8 | 62.6 |
| | + TentClf [ICLR'21] (Wang et al., 2021) | | 73.0±1.5 | 78.6±1.8 | 59.3±0.6 | 38.3±3.4 | 62.3 |
| | + T3A [NeurIPS'21] (Iwasawa & Matsuo, 2021) | | 77.3±1.5 | 80.8±0.7 | 63.2±0.5 | 40.2±0.6 | 65.4 |
| | + TAST [ICLR'23] (Jang & Chung, 2023) | | 77.3±0.7 | 81.9±0.4 | 63.7±0.5 | 42.6±0.7 | 66.4 |
| | + UniDG [Ours] | | **80.9 ± 0.1** | **81.7 ± 0.1** | **58.4 ± 0.1** | **47.9 ± 0.7** | **67.2 ↑ 0.8** |
| ERM (Vapnik, 1991) | + None | ResNet-50 | 76.7±0.5 | 83.2±1.1 | 67.1±1.0 | 45.9±1.3 | 68.3 |
| | + PL [ICMLW'13] (Lee, 2013) | | 69.4±3.1 | 81.7±4.6 | 62.9±3.1 | 38.1±2.4 | 63.0 |
| | + PLClf [ICMLW'13] (Lee, 2013) | | 75.7±0.9 | 83.3±1.6 | 67.0±1.0 | 46.7±2.1 | 68.2 |
| | + SHOT [ICML'20] (Liang et al., 2020) | | 67.1±0.9 | 84.1±1.2 | 67.7±0.7 | 35.2±0.8 | 63.5 |
| | + Tent [ICLR'21] (Wang et al., 2021) | | 73.0±1.3 | 85.2±0.6 | 66.3±0.8 | 37.1±2.0 | 65.4 |
| | + TentBN [ICLR'21] (Wang et al., 2021) | | 69.7±1.2 | 83.7±1.2 | 67.9±0.9 | 43.9±1.3 | 66.3 |
| | + TentClf [ICLR'21] (Wang et al., 2021) | | 75.8±0.7 | 82.7±1.6 | 66.8±1.0 | 43.6±2.6 | 67.2 |
| | + T3A [NeurIPS'21] (Iwasawa & Matsuo, 2021) | | 77.3±0.4 | 83.9±1.1 | 68.3±0.8 | 45.6±1.1 | 68.8 |
| | + TAST [ICLR'23] (Jang & Chung, 2023) | | 77.7±0.5 | 84.1±1.2 | 68.6±0.7 | 47.4±2.1 | 69.5 |
| | + UniDG [Ours] | | **81.6 ± 0.1** | **89.0 ± 0.3** | **68.9 ± 0.1** | **52.9 ± 0.2** | **73.1 ↑ 3.6** |

Table 3: Domain generalization accuracy with different backbone networks. UniDG improves the performance agnostic to visual backbones. **Bold** type indicates performance improvement.

| Type | Backbone | Method | VLCS | PACS | OfficeHome | Terra | Avg |
|---|---|---|---|---|---|---|---|
| Light-weight Networks | ResNet-18 (He et al., 2016) | ERM | 76.5 ± 0.1 | 79.2 ± 0.1 | 56.0 ± 0.1 | 40.3 ± 0.0 | 63.0 |
| | | + UniDG | **80.9 ± 0.1 ↑ 4.4** | **81.7 ± 0.1 ↑ 2.5** | **58.4 ± 0.1 ↑ 2.4** | **47.9 ± 0.7 ↑ 7.6** | **67.2 ↑ 4.2** |
| | MobilenetV3 (Howard et al., 2019) | ERM | 65.5 ± 0.2 | 79.1 ± 0.0 | 60.8 ± 0.2 | 30.4 ± 0.1 | 58.9 |
| | | + UniDG | **76.2 ± 0.1 ↑ 10.7** | **85.3 ± 0.4 ↑ 6.2** | **65.1 ± 0.2 ↑ 4.3** | **34.7 ± 0.2 ↑ 4.3** | **65.3 ↑ 6.4** |
| | EfficientNetV2 (Tan & Le, 2021) | ERM | 69.9 ± 0.2 | 89.2 ± 0.0 | 73.6 ± 0.2 | 36.0 ± 0.2 | 67.2 |
| | | + UniDG | **78.6 ± 0.2 ↑ 8.7** | **90.9 ± 0.1 ↑ 1.7** | **77.2 ± 0.1 ↑ 3.6** | **41.7 ± 0.4 ↑ 5.7** | **72.1 ↑ 4.9** |
| Convolution Networks | ResNet-50 (He et al., 2016) | ERM | 77.1 ± 0.1 | 82.9 ± 0.1 | 65.2 ± 0.1 | 45.4 ± 0.1 | 67.6 |
| | | + UniDG | **81.6 ± 0.1 ↑ 4.5** | **89.0 ± 0.3 ↑ 6.1** | **68.9 ± 0.1 ↑ 3.7** | **52.9 ± 0.2 ↑ 7.5** | **73.1 ↑ 5.5** |
| | ResNet-101 (He et al., 2016) | ERM | 76.4 ± 0.1 | 86.1 ± 0.0 | 67.4 ± 0.1 | 42.7 ± 0.1 | 68.1 |
| | | + UniDG | **80.5 ± 0.2 ↑ 4.1** | **88.3 ± 0.1 ↑ 2.2** | **70.3 ± 0.2 ↑ 2.9** | **50.0 ± 0.5 ↑ 7.3** | **72.3 ↑ 4.2** |
| | ConvNeXt-B (Liu et al., 2022) | ERM | 79.4 ± 0.0 | 92.7 ± 0.1 | 85.9 ± 0.1 | 60.9 ± 0.0 | 79.7 |
| | | + UniDG | **85.8 ± 0.3 ↑ 6.4** | **95.3 ± 0.2 ↑ 2.6** | **88.5 ± 0.1 ↑ 2.6** | **65.3 ± 0.3 ↑ 4.4** | **83.7 ↑ 4.0** |
| Transformer Networks | ViT-B16 (Dosovitskiy et al., 2020) | ERM | 78.4 ± 0.1 | 80.3 ± 0.1 | 75.6 ± 0.1 | 43.6 ± 0.0 | 69.5 |
| | | + UniDG | **83.6 ± 0.1 ↑ 5.0** | **85.4 ± 0.5 ↑ 5.1** | **81.0 ± 0.0 ↑ 5.4** | **51.4 ± 0.2 ↑ 8.0** | **75.4 ↑ 5.9** |
| | ViT-L16 (Dosovitskiy et al., 2020) | ERM | 76.4 ± 0.1 | 91.2 ± 0.1 | 83.3 ± 0.0 | 45.5 ± 0.0 | 74.1 |
| | | + UniDG | **83.2 ± 0.2 ↑ 6.8** | **95.2 ± 0.1 ↑ 4.0** | **87.5 ± 0.2 ↑ 4.2** | **53.9 ± 0.4 ↑ 8.4** | **79.9 ↑ 5.8** |
| | Hybrid ViT (Dosovitskiy et al., 2020) | ERM | 79.1 ± 0.1 | 89.1 ± 0.1 | 79.6 ± 0.1 | 52.9 ± 0.1 | 75.4 |
| | | + UniDG | **83.5 ± 0.1 ↑ 4.4** | **93.5 ± 0.1 ↑ 4.4** | **81.3 ± 0.1 ↑ 1.7** | **60.1 ± 0.4 ↑ 7.2** | **79.6 ↑ 4.2** |
| | DeiT (Touvron et al., 2021) | ERM | 79.5 ± 0.1 | 88.0 ± 0.1 | 77.0 ± 0.1 | 49.5 ± 0.1 | 73.5 |
| | | + UniDG | **85.1 ± 0.1 ↑ 5.6** | **92.6 ± 0.3 ↑ 4.7** | **79.5 ± 0.1 ↑ 2.5** | **54.1 ± 0.4 ↑ 4.8** | **77.8 ↑ 4.4** |
| | Swin Transformer (Liu et al., 2021b) | ERM | 80.0 ± 0.1 | 90.2 ± 0.1 | 81.6 ± 0.1 | 57.0 ± 0.0 | 77.2 |
| | | + UniDG | **85.0 ± 0.1 ↑ 5.0** | **94.3 ± 0.2 ↑ 4.1** | **84.6 ± 0.1 ↑ 3.0** | **62.0 ± 0.3 ↑ 5.0** | **81.5 ↑ 4.3** |
| Multi-Layer Perceptron | Mixer-B16 (Tolstikhin et al., 2021) | ERM | 73.6 ± 0.1 | 75.8 ± 0.0 | 52.4 ± 0.1 | 26.8 ± 0.1 | 57.2 |
| | | +UniDG | **81.3 ± 0.2 ↑ 7.7** | **82.3 ± 0.1 ↑ 6.5** | **57.7 ± 0.3 ↑ 5.2** | **41.2 ± 0.5 ↑ 14.4** | **65.6 ↑ 8.4** |
| | Mixer-L16 (Tolstikhin et al., 2021) | ERM | 77.1 ± 0.1 | 85.0 ± 0.1 | 70.3 ± 0.1 | 36.6 ± 0.0 | 67.4 |
| | | + UniDG | **83.0 ± 0.1 ↑ 4.9** | **88.5 ± 0.2 ↑ 3.5** | **75.6 ± 0.1 ↑ 5.3** | **45.0 ± 1.4 ↑ 8.4** | **73.0 ↑ 5.6** |

### 3.2.2 TEST-TIME ADAPTATION BENCHMARKS

UniDG remarkably outperforms all existing test-time methods including the state-of-the-art method, TAST (Jang & Chung, 2023). Specifically, as shown in Table 2, we choose ResNet-18 and ResNet-50 as the backbone and average accuracy as the metric to evaluate several test-time methods. UniDG achieves an average accuracy of 67.2% with ResNet-18 on VLCS, PACS, OfficeHome, and terrain, which is 0.8% higher than the best-performing test-time method. The superiority of UniDG is even more significant with ResNet-50: UniDG achieves an average accuracy of 73.1% on four benchmarks, largely exceeding the last state of the art, TAST (Jang & Chung, 2023), by 3.5%.

Except for ResNet-18 and ResNet-50, we further use UniDG with 12 mainstream backbones including CNN, MLP, and transformer architectures and report the results in Figure 3. It turns out that UniDG can significantly improve the performance of all the 12 backbones so that we conclude UniDG is a universal architecture-agnostic method. Notably, the number of parameters of these models ranges from 1.59M to 303M, but UniDG can significantly and consistently improve the performance by 5.4% on average.

### 3.3 ABLATION STUDY

**Effectiveness of Marginal Generalization.** Table 4 shows Marginal Generalization significantly improves the performance on target domains compared with the baseline model

Figure 5: Accuracy accumulation curves on VLCS. UniDG outperforms the base ERM model by about 5% in accuracy. Note we randomly select **10 different trial seeds** for better comparison.

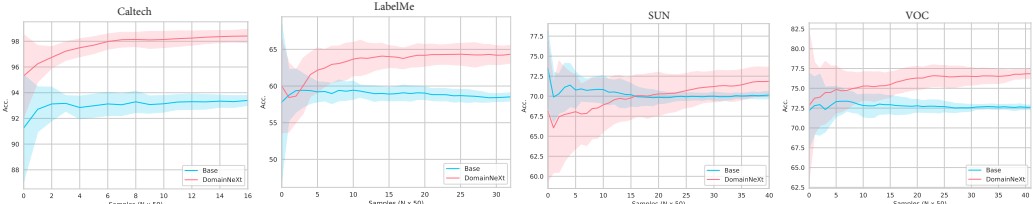

by +3.3% (70.9% vs. 67.6%). With the classifier adaptation scheme (§ 2.3) but no Marginal Generalization, the performance reaches 70.8%, bringing a +3.2% improvement. While further integrating the two schemes, the ability of the network for domain generalization gets significantly boosted, increasing from 67.6% to 71.9%.

**Effectiveness of Differentiable Memory Bank** As shown in the 4th, 5th, and 7th rows of Table 4, Differentiable Memory Bank (§ 2.3) also significantly improves the generalization ability of the model. Referring to the 4th row, the memory bank effectively boosts the performance of the base model from 67.6% to 70.4% (+2.8%). Meanwhile, when combining differentiable memory bank and Marginal Generalization, a further improvement of +5.5% can be achieved. It reveals that the proposed schemes can be mutually beneficial, where the adapted model

Table 4: **Ablation Study**. We take the mean accuracy (mAcc) on the PACS, VLCS, OfficeHome, and TerraInc datasets as the evaluation metric.

| Ablation Components | | | | mAcc | Δ |
|---|---|---|---|---|---|
| $\mathcal{L}_m$ (Eq. 5) | $\mathcal{L}_e$ (Eq. 6) | $\mathcal{M}$ | $\omega \leftarrow v$ | (%) | (%) |
| ✗ | ✗ | ✗ | ✗ | $67.6 \pm 0.1$ | − |
| ✓ | ✗ | ✗ | ✗ | $70.9 \pm 0.2$ | **+3.3** |
| ✗ | ✓ | ✗ | ✗ | $70.8 \pm 0.1$ | **+3.2** |
| ✗ | ✗ | ✓ | ✗ | $70.4 \pm 0.2$ | **+2.8** |
| ✗ | ✗ | ✗ | ✓ | $70.7 \pm 0.1$ | **+3.1** |
| ✓ | ✓ | ✗ | ✗ | $71.9 \pm 0.3$ | **+4.3** |
| ✗ | ✗ | ✓ | ✓ | $71.6 \pm 0.0$ | **+4.0** |
| ✓ | ✓ | ✓ | ✓ | $73.1 \pm 0.2$ | **+5.5** |

has refined gradients and a differentiable memory bank receives better prototypes. Thus they enhance the ability of networks to generalization together.

## 3.4 QUANTITATIVE ANALYSIS

Figure 5 shows the accumulation curves of each instance interval across four domains on the VLCS (Li et al., 2017) dataset by 10 parallel trials. UniDG brings significant and stable improvements on each domain, for which the fluctuation range of accumulation accuracy is close to the base model and mean scores are prominently improved.

Table 5: Source knowledge preserve and training efficiency of UniDG.

(a) Source Knowledge Preserve

| VLCS | L | S | V |
|---|---|---|---|
| Source model | 96.02 | 97.14 | 98.33 |
| TENT | 92.15 −3.9 | 94.23 −2.9 | 95.13 −3.2 |
| UniDG | 94.32 −1.7 | 96.68 −0.4 | 97.63 −0.7 |

(b) Efficiency of UniDG

| Method | Wall Clock Time (s) |
|---|---|
| TENT (Res50) | 0.581 |
| UniDG (Res50) | 0.587 |

As shown in Table 5, 1) referring to Table 5a, we observe a smaller performance decrease of UniDG after adaption on the source domains. It proves that UniDG can better preserve pretrained source knowledge. 2) In Table 5b, we detail the training efficiency of UniDG and compare our method with TENT on wall clock time with the NVIDIA A100 GPU. It reveals that although we propose to update the parameters of the whole network, the computation burden will not sharply increase.

## 3.5 COMMONALITY ANALYSIS

1) **Light-weight Networks** UniDG brings out significant average improvements of 5.1% on light-weight MobileNet V3 (Howard et al., 2019), EfficientNet V2 (Tan & Le, 2021), and ResNet 18 (He et al., 2016). For example, the accuracy of MobileNet V3 has been improved by as much as 6.4%, which proves the strong feasibility of UniDG to improve the performance of edge devices for generalizing in unseen environments. 2) **Architecture-Free** UniDG is a unified solution based on online adaptation to handle domain shifts. As shown in Table 3, UniDG has a general improvement of about 5% on 10+ mainstream visual networks including CNN, Transformer, and MLP as their back-

bones. The highest improvement comes from Mixer-B16 (Tolstikhin et al., 2021), which increased from 57.2% to 65.6%.

## 4 RELATED WORK

**Domain Generalization** Domain Generalization (DG) can be classified into three types: 1) **Representation Learning**: These methods extract specific features from source domains and assume them robust in target domains. One approach is domain alignment (Li et al., 2018c;b; Matsuura & Harada, 2020), extracting domain-invariant representations from source domains, which is a non-trivial task. Therefore feature disentanglement (Rojas-Carulla et al., 2018; Piratla et al., 2020; Christiansen et al., 2021; Mahajan et al., 2021; Sun et al., 2021; Liu et al., 2021a), loosens the constraint, learning disentangled representations. 2) **Foundation Models**: different backbones reveal the diverse ability to tackle the DG problem. These methods (Li et al., 2017; Ding & Fu, 2017; Carlucci et al., 2019; Li et al., 2023) optimize the architecture of the mainstream backbone for DG. GMoE (Li et al., 2023) based on ViT (Dosovitskiy et al., 2020), replaces the FFN layers with mixture-of-experts, allowing different experts to focus on the different visual attributes. 3) **Learning Strategy**: These methods utilize machine learning strategy to enhance the model's generalization capability on target domains, including meta-learning and ensemble learning. Meta-learning (Li et al., 2018a; 2019b;a; Dou et al., 2019; Liu et al., 2020; Chen et al., 2022d; Li et al., 2021) divide training data into meta-train and meta-test sets, then simulate domain shift and update parameters during training. Ensemble learning (Ding & Fu, 2017; Zhou et al., 2021; Cha et al., 2021) learns model copies to extract features and migrate their ensemble to target domains.

**Continual Learning** Continual learning (De Lange et al., 2021) aims to relieve continuous domain shifts, which face complicated catastrophic forgetting. Existing methods (Rebuffi et al., 2017; Zenke et al., 2017; Kirkpatrick et al., 2017; Li & Hoiem, 2017; Lao et al., 2020) propose regularization and replay to reinforce learning representations space from parameters and data stream perspectives. Recently,self-supervised learning (Radford et al., 2015; He et al., 2022; Grill et al., 2020) utilize prior knowledge obtained by pre-training with massive datasets and have shown strong performance in DG. Radford et al. (2021) trains image encoder and text encoder jointly, matching 400 million (image, text)pairs. Besides, researchers have noted the superiority of causal learning (Zhou et al., 2021; Mahajan et al., 2021) in domain generalization.

**Test-time Adaptation** TTA schemes (Karani et al., 2021; Iwasawa & Matsuo, 2021; Sun et al., 2020; Park et al., 2023) propose to update model parameters based on target data.

1) **Adversarial Learning**: With the advancement of generative adversarial networks, Li et al. (2020); Yeh et al. (2021); Kurmi et al. (2021) generate target data with generative models, improving the ability to handle domain shift without the support of source data.2) **normalization-based**: The normalization method replaces the batch normalization (BN) statistics of the trained model with the BN statistics estimated on test data and updates parameters of the BN layers only, with the backbone network frozen. Wang et al. (2021) aims to minimize entropy during testing. Schneider et al. (2020) uses Wasserstein distance between source and target statistics as the measurement. 3) **Bayesian Learning**: Zhou & Levine (2021) analyses TTA from a Bayesian perspective (Li et al., 2016; Hu et al., 2021; You et al., 2021) and proposes a regularized entropy minimization procedure achieved by approximating the probability density during the training time.

## 5 DISCUSSION AND CONCLUSION

Aiming at the OOD problem, this paper proposes a general self-supervised online learning scheme, named UniDG , to update all the parameters of the model during the testing phase. Specifically, UniDG contains Marginal Generalization and Differentiable Memory Bank, which can successfully balance the conservation of source knowledge and generalization ability to novel environments. Our method shows high effectiveness and potential for complex domain shifts in actual scenarios. On four domain generalization benchmarks, UniDG achieved a new state-of-the-art performance with an average accuracy of 79.6%. Additionally, UniDG improved 12 backbone models by an average of 5.4%. By comparing with existing pre-trained model and other test-time methods, we show it is a promising direction to develop the online adaptation method to deal with the OOD problem.

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

# Appendix

## A  SUMMARY

This appendix describes more details of the ICLR 2024 submission, titled *Towards Unified and Efficient Domain Generalization*. The appendix is organized as follows:

- § B theoretically discusses why UniDG can perform better under the setting of test-time adaptation than the BN-based approaches.
- § C provides more implementation details and analytical experiment for the hyper-parameter $\sigma$ of Equation 5.
- § D summarizes the pseudo-code of our proposed UniDG .
- § E illustrates the effectiveness of the proposed UniDG by quantities of T-SNE visualization results compared with existing advanced methods including TENT Wang et al. (2021) and TAST Jang & Chung (2023).
- § F exhibits a more detailed comparison between the other test-time schemes and provides improvements of each visual backbone on all domains.

## B  THEORETICAL INSIGHT

To mitigate catastrophic forgetting during test-time adaptation, existing methods such as TENT Wang et al. (2021), SHOT Liang et al. (2020), and TAST Jang & Chung (2023) propose adapting the parameters of Batch Normalization (BN) layers. We argue that adapting only the BN layers may be insufficient for effectively handling the unseen novel domains compared to adapting the entire network parameters. To substantiate our claim, we provide a theoretical analysis from two perspectives:

1) Neural Tangent Kernels (NTK) Jacot et al. (2018): NTK is a kernel that elucidates the evolution of neural networks during training via gradient descent. It bridges the gap between neural networks and classical kernel methods in machine learning. When the width of the hidden layers in a neural network approaches infinity, the network's training behavior becomes more predictable. In this paper, we employ neural tangent kernels to assess the network's ability to generalize to unseen domains $x^T \in \mathcal{T}$ while training on the source domain $x^S \in \mathcal{S}$.

2) Gradient descent process of BN layers restricts the expansion of neural tangent kernels: We argue that solely adapting the BN layers could limit the growth of neural tangent kernels, affecting the model's generalization capability for unseen domains.

In summary, our theoretical analysis highlights the potential limitations of adapting only the BN layers in handling novel domains and suggests that a more comprehensive approach might be necessary to achieve better generalization.

### B.1  NEURAL TANGENT KERNEL IN DG

For domain generalization, the networks can be formulated from two parts: feature extractor $f(\cdot)$ and classifier $q(\cdot)$. And Neural Tangent Kernel $K_{\cdot,\cdot}$ formulates the impact of gradients between different instances $x^S, x^T$ to the learning representations of the neural network in the gradient decent progress:

$$K_{x^S, x^T} = k(x^S, x^T) = \lim_{\eta \to 0} \frac{f(x, \theta + \eta[\frac{\nabla f_\theta(x)}{\nabla \theta}] - f(x, \theta))}{\eta},\qquad(9)$$

where $\eta$ is the learning rate and $\theta$ denotes the parameters. Accordingly, we can obtain the relationship between parameters and learning representations with learning rate $\eta$ and neural tangent kernel $K_{x^s x^T}$ on the source and target domains:

$$f(x^\top, \theta + \eta[\frac{\nabla f(x^T)}{\eta\theta}]) = k(x^S, x^T)\eta + f(x^S, \theta) - f(x^T, \theta) = K_{x^S x^T} \cdot \eta + f(x^S, \theta) - f(x^T, \theta).$$
$$(10)$$

Meanwhile, we can also obtain the formula of neural tangent kernels according to its definition Jacot et al. (2018):

$$K_{x^S x^T} = \mathbb{E}_{\theta \sim \omega}[f(x^S, \theta) \cdot f(x^T, \theta)] \quad = \mathbb{E}_{\theta \sim \omega}[\langle \frac{\partial f(x^S, \theta)}{\partial \theta}, \frac{\partial f(x^T, \theta)}{\partial \theta} \rangle], \tag{11}$$

where $\mathbb{E}_{\theta \sim \omega}(\cdot)$ indicates the mathematical expectation of network parameter in the parameters space $\omega$, and $\frac{\partial f(x^S, \theta)}{\partial \theta}$ and $\frac{\partial f(x^T, \theta)}{\partial \theta}$ denotes the gradients of the parameters with representation on the source and target domains, respectively. And the neural tangent kernels in domain generalization can be regarded as the inner product of the gradients for learning representations on the source and target domains.

## B.2 BACKWARD GRADIENTS IN BN LAYERS

Batch Normalization operation is focused on introducing means $\mu$ and standard error $\sigma^2$ to normalize learning representations in a mini-batch $\mathcal{B} = \{x_1, x_2, \ldots, x_m\}$ containing $m$ samples. And BN layer introduces linear projection with two learnable parameters $\gamma$ and $\beta$ based on the Batch Normalization operation. Its computation can be formulated as:

$$\mu_\mathcal{B} \leftarrow \frac{1}{m} \sum_{i=1}^{m} x_i, \ \sigma_\mathcal{B}^2 \leftarrow \frac{1}{m} \sum_{i=1}^{m} (x_i - \mu_B)^2, \ \hat{x}_i \leftarrow \frac{x_i - \mu_\beta}{\sqrt{\sigma_\beta^2 + \varepsilon}}, \ \hat{y}_i \leftarrow \gamma \hat{x}_i + \beta, \tag{12}$$

where $\hat{y}_i$ is the output of BN layer, which can be simplified as: $y_i = \text{BN}_{\gamma, \beta}(x_i)$, $\varepsilon$ is a value to smooth the computation. And we can obtain the formula to describe it with involved variables including $m, \gamma, \beta, \sigma, x_i$:

$$\hat{y}_i = \gamma \frac{x_i - \frac{1}{m} \sum_{i=1}^{m} x_i}{\sqrt{\frac{1}{m} \sum_{i=1}^{m} (x_i - \frac{1}{m} \sum_{i=1}^{m} x_i)^2 + \varepsilon}}. \tag{13}$$

According to the Equation 13, we can utilize chain rules to calculate the gradients between representations and input with intermediate variables:

$$\frac{\partial \hat{x}_i}{\partial \mu} = -\frac{1}{\sqrt{\sigma^2 + \varepsilon}}, \ \frac{\partial \hat{x}}{\partial \sigma^2} = \sum_{i=1}^{m} (x_i - \mu)^{-\frac{1}{2}}, (\sigma^2 + \epsilon)^{-\frac{3}{2}}, \ \frac{\partial \sigma^2}{\partial \mu} = \frac{1}{m} \sum_{i=1}^{m} -(x_i - \mu). \tag{14}$$

Furthermore, we can derive the gradient of learning representation $f$ in BN layers according to Equation 12 for learnable parameters $\gamma, \beta$:

$$\frac{\partial f}{\partial \gamma} = \frac{\partial f}{\partial y_i} \cdot \frac{\partial y_i}{\partial \gamma} = \sum_{i=1}^{m} \frac{\partial f}{\partial y_i} \cdot \hat{x}, \quad \frac{\partial f}{\partial \beta} = \frac{\partial f}{\partial y_i} \cdot \frac{\partial y}{\partial \beta} = \sum_{i=1}^{m} \frac{\partial f}{\partial y_i}. \tag{15}$$

And we can also obtain the gradients for representations and statistic variable in the mini-batch by chain rules:

$$\frac{\partial f}{\partial \hat{x}_i} = \frac{\partial f}{\partial y} \cdot \frac{\partial y_i}{\partial x_i} = \frac{\partial f}{\partial y_i} \cdot \gamma, \quad \frac{\partial f}{\partial \sigma^2} = \frac{\partial f}{\partial \hat{x}} \cdot \frac{\partial \hat{x}}{\partial \sigma^2}, \quad \frac{\partial f}{\partial u} = \frac{\partial f}{\partial \hat{x}_i} \frac{\partial \hat{x}_i}{\partial \mu} + \frac{\partial f}{\partial \sigma^2} \frac{\partial \sigma^2}{\partial \mu}, \tag{16}$$

and we have obtained $\frac{\partial \hat{x}_i}{\partial \mu}$ and $\frac{\partial \hat{x}}{\partial \sigma^2}$ in Equation 14. Then, for the unknown gradients of $\frac{\partial f}{\partial \mu}$, we can utilize the results in Equation 14 to simplify:

$$
\begin{aligned}
\frac{\partial f}{\partial \mu} &= (\sum_{i=1}^{m} \frac{\partial f}{\partial \hat{x}_i} \cdot \frac{-1}{\sqrt{\sigma^2 + \varepsilon}}) + (\frac{\partial f}{\partial \sigma^2} \cdot \frac{1}{m} \sum_{i=1}^{m} -2(x_i - \mu)), \\
&= \left( \sum_{i=1}^{m} \frac{\partial f}{\partial \hat{x}_i} - \frac{-1}{\sqrt{\sigma^2 + \varepsilon}} \right) + \left( \frac{\partial f}{\partial \sigma^2} \cdot \left( \frac{-2}{m} \sum_{i=1}^{m} x_i + \frac{2}{m} \sum_{i=1}^{m} u \right) \right), \\
&= \left( \sum_{i=1}^{m} \frac{\partial f}{\partial \hat{x}_i} \cdot \frac{-1}{\sqrt{\sigma^2 + \varepsilon}} \right) + \left( \frac{\partial f}{\partial \sigma^2} \cdot \left( -2u + \frac{2m \cdot u}{m} \right) \right) = \sum_{i=1}^{m} \frac{\partial f}{\partial \hat{x}_i} \cdot \frac{-1}{\sqrt{\sigma^2 + \varepsilon}},
\end{aligned}
\tag{17}
$$

In addition, the gradients of $\frac{\partial f}{\partial x_i}$ is directly calculated with chain rules as: $\frac{\partial f}{\partial x_i} = \frac{\partial f}{\partial \hat{x}_i} \cdot \frac{\partial \hat{x}_i}{\partial x_i} + \frac{\partial f}{\partial u} + \frac{\partial u}{\partial x_i} + \frac{\partial f}{\partial \sigma^2} \cdot \frac{\partial \sigma^2}{\partial x_i}$. Referring to Equation 12, these gradients can be simply obtained $\frac{\partial \hat{x}_i}{\partial x_i} = \frac{1}{\sqrt{\sigma^2 + \varepsilon}}$, $\frac{\partial u}{\partial x_i} = \frac{1}{m}$, and $\frac{\partial \sigma^2}{\partial x_i} = \frac{2(x_i - u)}{m}$. Therefore, we can compute the gradient $\frac{\partial f}{\partial x_i}$ as:

$$\frac{\partial f}{\partial x_i} = \left( \frac{\partial f}{\partial \hat{x}_i} \cdot \frac{1}{\sqrt{\sigma^2 + \varepsilon}} \right) + \left( \frac{\partial f}{\partial u} \cdot \frac{1}{m} \right) + \left( \frac{\partial f}{\partial \sigma^2} \cdot \frac{2(x_i - u)}{m} \right), \tag{18}$$

According to Equation 14, 16, 17, we compute the gradient as:

$$\begin{aligned}
\frac{\partial f}{\partial x_i} &= \left( \frac{\partial f}{\partial \hat{x}_i} \cdot \frac{1}{\sqrt{\sigma^2 + \varepsilon}} \right) + \left( \frac{\partial f}{\partial u} \cdot \frac{1}{m} \right) + \left( \frac{\partial f}{\partial \sigma^2} \cdot \frac{2(x_i - u)}{m} \right) \\
&= \left( \frac{\partial f}{\partial \hat{x}_i} \cdot \frac{1}{\sqrt{\sigma^2 + \varepsilon}} \right) + \left( \frac{1}{m} \sum_{j=1}^m \frac{\partial f}{\partial \hat{x}_j} \cdot \frac{-1}{\sqrt{\sigma^2 + \varepsilon}} \right) - \left( \frac{1}{2} \sum_{j=1}^m \frac{\partial f}{\partial \hat{x}} (x_j - u) \cdot (\sigma^2 + \varepsilon)^{-\frac{3}{2}} \cdot \frac{2(x_i - u)}{m} \right) \\
&= \left( \frac{\partial f}{\partial \hat{x}} \cdot (\sigma^2 + \varepsilon)^{-\frac{1}{2}} \right) - \left( \frac{(\sigma^2 + \varepsilon)^{-\frac{1}{2}}}{m} \cdot \sum_{j=1}^m \frac{\partial f}{\partial \hat{x}_j} \right) + \left( \frac{(\sigma^2 + \varepsilon)^{-\frac{1}{2}}}{m} \cdot \hat{x}_i \cdot \sum_{j=1}^m \frac{\partial f}{\partial \hat{x}_j} \cdot \hat{x}_j \right) \\
&= \frac{(\sigma^2 + \varepsilon)^{-\frac{1}{2}}}{m} \left[ m \frac{\partial f}{\partial \hat{x}_i} - \sum_{j=1}^m \frac{\partial f}{\partial \hat{x}_j} - \hat{x}_i \sum_{\hat{v}=1}^m \frac{\partial f}{\partial \hat{x}_j} \cdot \hat{x}_j \right]
\end{aligned} \tag{19}$$

Therefore, refer to Equation 16, we can obtain the gradients $\frac{\partial f}{\partial \hat{x}_i}$ related to $\frac{\partial f}{\partial y_i}$, which can be directly calculated with known gradients including $\frac{\partial f}{\partial \hat{x}_i} = \frac{\partial f}{\partial y_i} \cdot \gamma$, $\frac{\partial f}{\partial \beta} = \sum_{i=1}^m \frac{\partial f}{\partial y_i}$, $\frac{\partial f}{\partial \gamma} = \sum_{i=1}^m \frac{\partial f}{\partial y_i} \cdot \hat{x}_i$. Finally, the backward gradients in the BN layer can be computed as:

$$\frac{\partial f}{\partial x_i} = \frac{m \frac{\partial f}{\partial \hat{x}_i} - \sum_{j=1}^m \frac{\partial f}{\partial \hat{x}_j} - \hat{x}_i \sum_{j=1}^m \frac{\partial f}{\partial \hat{x}_j} \cdot x_j}{m\sqrt{\sigma^2 + \varepsilon}} \tag{20}$$

According to results in Equation 9 and 20, we can describe the adaptation process for these approaches adapting `BN` layers:

$$\begin{aligned}
K_{x^S, x^T} &= E_{\theta \sim W} \left[ < \frac{\partial f(x^S, \theta)}{\partial \theta}, \frac{\partial f(x^T, \theta)}{\partial \theta} > \right] \\
\frac{\partial \text{BN}(x)}{\partial x_i} &= \frac{m \frac{\partial f}{\partial \hat{x}_i} - \sum_{j=1}^m \frac{\partial f}{\partial \hat{x}_j} - \hat{x}_i \sum_{j=1}^m \frac{\partial f}{\partial \hat{x}_j} \hat{x}_j}{m\sqrt{\sigma^2 + \varepsilon}} \\
K^{\text{BN}}_{x^S, x^T} &= \frac{1}{m} \sum_{i=1}^m [(\frac{\partial \text{BN}(x^T)}{\partial x_i^T} \cdot \frac{\partial \text{BN}(x^S)}{\partial x_i^S}) / (\|\frac{\partial \text{BN}(x^T)}{\partial x_i^T}\| \cdot \|\frac{\partial \text{BN}(x^S)}{\partial x_i^S}\|)] \\
&= \frac{1}{m} \sum_{i=1}^m [\frac{m \frac{\partial f}{\partial \hat{x}_i^T} - \sum_{j=1}^m \frac{\partial f}{\partial \hat{x}_j^T} - \hat{x}_i^T \sum_{j=1}^m \frac{\partial f}{\partial \hat{x}_j^T} \hat{x}_j^T}{m\sqrt{\sigma^2 + \varepsilon}}] \cdot [\frac{m \frac{\partial f}{\partial \hat{x}_i^S} - \sum_{j=1}^m \frac{\partial f}{\partial \hat{x}_j^S} - \hat{x}_i^S \sum_{j=1}^m \frac{\partial f}{\partial \hat{x}_j^S} \hat{x}_j^S}{m\sqrt{\sigma^2 + \varepsilon}}] \\
&\quad / (\|\frac{\partial \text{BN}(x^T)}{\partial x_i^T}\| \cdot \|\frac{\partial \text{BN}(x^S)}{\partial x_i^S}\|) \\
&\leq \frac{1}{(\sigma^2 + \varepsilon)} [\frac{\partial f}{\partial \hat{x}_i^S} - \frac{\partial f}{\partial \hat{x}_j^S} - \hat{x_i^S} \cdot \frac{\partial f}{\partial \hat{x}_j^S} \hat{x_j^S}]^2 / (\|\frac{\partial \text{BN}(x^S)}{\partial x_i^S}\|^2) \leq K_{x^S, x^S}
\end{aligned} \tag{21}$$

### B.3 CONCLUSION

Referring to Equation 21, it illustrates that **neural tangent kernels for networks pretrained on source has less width to get transferred on target domains if the algorithm is focused on adapting parameters of the `BN` layers.** Therefore, we propose to adapt the comprehensive parameter space to enhance the ability to generalize on the novel environments.

## C IMPLEMENTATION DETAILS

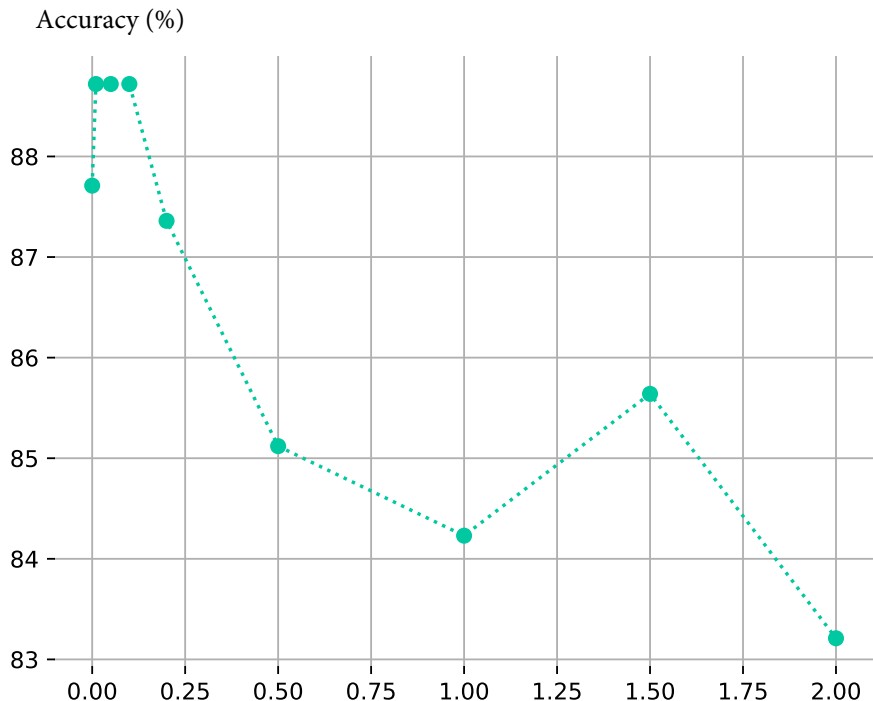

Figure 6: Hyper-parameter impact of $\sigma$.

We split the dataset into training and validation sets, where 80% of the samples are randomly selected as the training data and the rest 20% is the validation data. The validation set is exploited to select the hyper-parameter with the criterion of maximizing the accuracy of the validation set.

In the test-time adaptation setting, the adaptation starts with a trained source model, which will be trained using source data in advance. To build a test-time adaptation method for domain generalization (DG), we first acquired a source model with data from multiple source domains. In addition, two pre-training algorithms are exploited for obtaining the source models, i.e., ERM and CORAL. The experiment results are the averages of three trials with different random seeds. We train the source model using an Adam optimizer with a learning rate of $5e^{-5}$ and a batch size of 32. The learning rate for adaptation phase is varying $\in \{5e^{-4}, 5e^{-5}, 5e^{-6}\}$.

In Figure 6, we visualize the sensitivity of hyper-parameter $\sigma$ in Equation 5 on the PACS Torralba & Efros (2011) dataset. We take hyper-parameter $\sigma \in \{0, 0.01, 0.05, 0.10, 0.15, 0.20, 0.50, 1.0, 1.5, 2.0\}$. It is also worth noting that when $\sigma$ is set in $[0.01, 0.05, 0.15, 0.20]$, the network achieves better performances on the PACS dataset. Therefore, we choose to set $\sigma = 0.15$.

## D    PSEUDO CODE FOR UNIDG

For feature extractor $f$, we detach the gradient and freeze the parameters of the trained source model, and utilize it to extract the representation of images from target domains as the source knowledge. And we initialize another new network $\theta'$ with the trained parameters of $\theta$ and formulate the representation discrepancy between $f(\boldsymbol{x})$ and $f'(\boldsymbol{x})$ via Frobenius norm $\|\cdot\|_F$:

$$\mathcal{L}_m = \frac{1}{\|D_T\|} \sum_{i=1}^{\|D_T\|} [\|f'(\boldsymbol{x}_i) - f(\boldsymbol{x}_i)\|_F^2 - \sigma]. \tag{22}$$

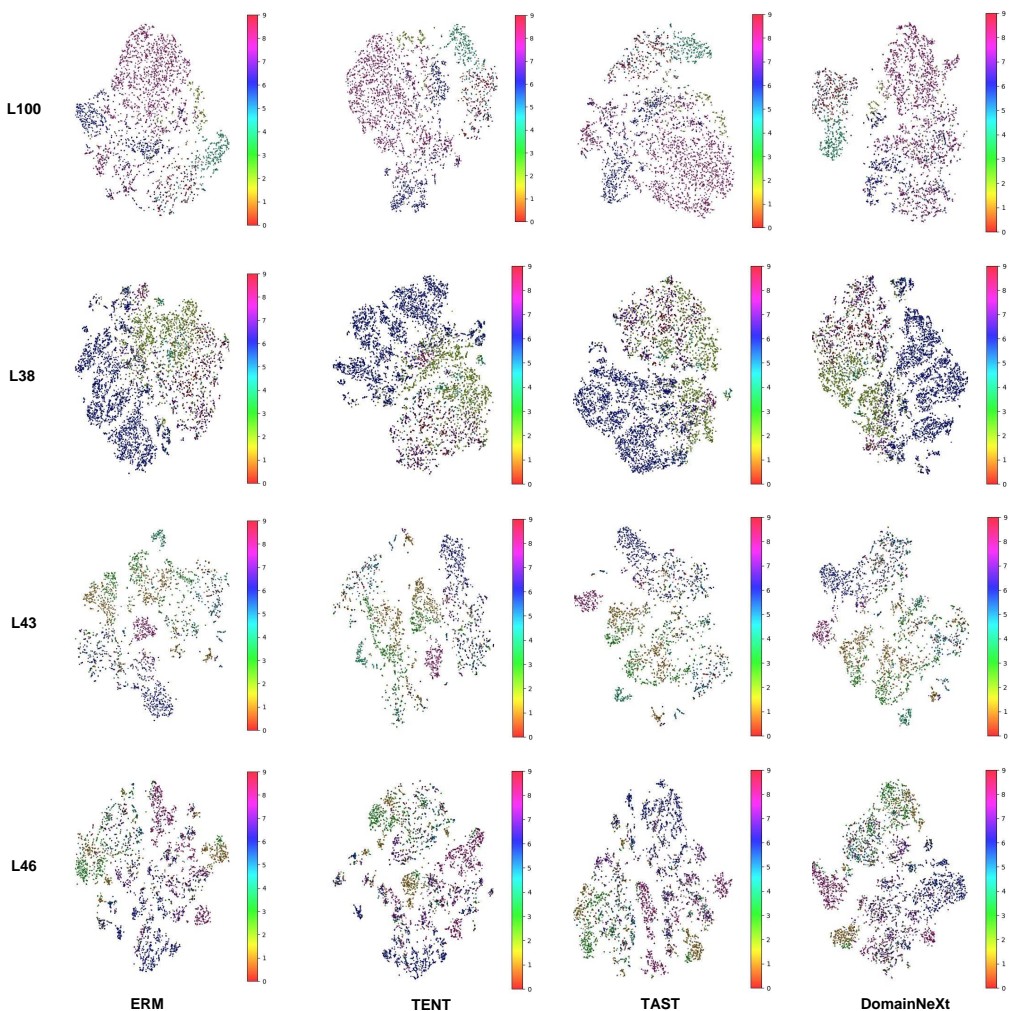

Figure 7: Qualitative Results of UniDG on challenging TerraIncognita Beery et al. (2018) dataset. UniDG (4th column) aggregates intra-class feature embeddings better with the pretrained ERM model (1st column) compared with advanced methods including TENT Wang et al. (2021) (2nd column), and TAST Jang & Chung (2023) (3rd column).

For representations $f'(\boldsymbol{x})$, we utilize a linear layer to work as a classifier and obtain the classification results $q'(f'(\boldsymbol{x}))$. Then we take the softmax entropy as the loss function to update the classifier:

$$\mathcal{L}_e = -\frac{1}{N_b} \sum\nolimits_{i=1}^{N_b} \log\left[\frac{\exp(q'(f'(\boldsymbol{x}_i)))}{\sum \exp q'(f'(\boldsymbol{x}))}\right]. \tag{23}$$

The memory bank is set to store the class-wise prototypes of each class $\mathcal{M} = \bigcup_{i=1}^{c}\{\boldsymbol{p}_i\}, \boldsymbol{p}_i \in \mathbb{R}^C$, where $\mathcal{M}$, $c$, and $C$ denote the memory bank, the number of classes, and feature dimensions. For each class $i$, the prototype $\boldsymbol{p}_i$ is initialized with parameters of classifier $\boldsymbol{p}_i arrow q'(\cdot)_i$:

$$\mathcal{M} = \bigcup_{i=1}^{c}\left[\frac{1}{K} \sum_{j=1}^{K} f'(\boldsymbol{x}_{\boldsymbol{j}}^{\boldsymbol{i}})\right]. \tag{24}$$

Besides utilizing the Top-$K$ re-ranking approach to select the prototypes, to further exploit the potential of the test-time scheme, we make the memory iteration learnable, which directly optimizes these prototypes via gradient descent. We construct the learnable memory terms by introducing

matrix products of learning representations $f'(\boldsymbol{x}_i)$, prototypes $\boldsymbol{p}_i$, and pseudo labels $\hat{y}_i$:

$$\gamma_i = f'(\boldsymbol{x}_i) \cdot \text{BN}(\frac{\boldsymbol{p}_i}{|\boldsymbol{p}_i|} \cdot \hat{y}_i),$$

$$\mathcal{L}_i = -\frac{1}{N_b} \sum_{i=1}^{N_b} \gamma_i \log[\frac{\exp(\gamma_i)}{\sum \exp(\gamma_i)}].$$

(25)

Therefore, our algorithm can be summarized as follows:

---

**Algorithm 1** UniDG Algorithm

---

**Require:** number of steps $T$; feature extractor $f$ and classifier $q$ pretrained from source domains; the unlabeled target domain.

1: $s = 0$
2: Copy a new model with the source one: $f' \leftarrow f, q' \leftarrow q$ and freeze $f, q$.
3: **for** $s \leq T$ **do**
4:   $s = s + 1$
5:   Arrived target data $\boldsymbol{x}_t$
6:   Update Prototypes $\boldsymbol{p}$ in Eq. 24 using $\boldsymbol{x}_t$
7:   Calculate the loss $\mathcal{L}_m$ using Eq. 5
8:   Calculate the loss $\mathcal{L}_e$ according to Eq. 23.
9:   Calculate the overall loss: $\mathcal{L} \leftarrow \mathcal{L}_i + \mathcal{L}_e + \mathcal{L}_m$
10:   Update the network models $f', q'$ using $\mathcal{L}$ in step 10.
11: **end for**

---

## E    VISUALIZATION RESULTS

In Figure 7, we visualize feature embeddings of ERM Vapnik (1998), TENT Wang et al. (2021), TAST Jang & Chung (2023), and proposed UniDG . Referring to the figure, we find that, feature embeddings extracted by UniDG have better intra-class compactness and inter-class separability.

## F    DETAILED RESULTS

In this section, we provide more detailed experimental results. These results are organized into three parts:

- We provide experiments of transferring BN-based methods to adapt LN layers methods on VLCS dataset in Table 6.
- Table 7-Table 10 demonstrate the comparison between existing test-time methods with ResNet-18 He et al. (2016) backbones and UniDG on the VLCS Li et al. (2017), PACS Torralba & Efros (2011), Office Home Venkateswara et al. (2017), and TerraIncognita Beery et al. (2018) datasets.
- Table 11-Table 14 show the comparison between UniDG and the other test-time methods with the ResNet-50 He et al. (2016) visual backbones.
- Table 15-Table 18 detail improvements on each domain between base ERM Vapnik (1998) models with different 12 visual backbones on the aforementioned 4 datasets.

Table 6: Comparsion between transferring existing BN-based method and our proposed UniDG with ViT-B16 backbone on the VLCS dataset.

| Method | C | L | S | V | Avg |
|---|---|---|---|---|---|
| Source | 98.50 | 63.44 | 74.79 | 77.79 | 78.63 |
| TENT (Wang et al., 2021) | 99.23 | 64.36 | 75.23 | 77.94 | 79.19 |
| UniDG (LN) | 99.73 | 66.65 | 78.44 | 79.79 | **81.15** ↑ 2.52 |
| UniDG | 99.91 | 69.60 | 82.86 | 82.82 | **83.80** ↑ 4.61 |

Table 7: Full results using classifiers trained by ERM for Table 2 on VLCS. We use ResNet-18 as a backbone network.

| Method | C | L | S | V | Avg |
|---|---|---|---|---|---|
| ERM | 94.70±1.33 | 63.79±1.30 | 67.90±1.97 | 73.15±1.37 | 74.88 |
| +Tent | 89.82±2.89 | 61.98±1.10 | 65.51±1.91 | 74.21±1.61 | 72.88 |
| +TentBN | 79.80±4.74 | 58.51±1.44 | 61.62±0.92 | 68.14±1.74 | 67.02 |
| +TentClf | 94.75±1.43 | 63.74±1.41 | 67.92±2.22 | 65.40±6.91 | 72.96 |
| +SHOT | 91.45±6.83 | 48.26±1.77 | 54.75±2.59 | 66.51±1.25 | 65.24 |
| +SHOTIM | 90.28±7.00 | 47.96±1.45 | 54.66±2.47 | 66.52±1.19 | 64.86 |
| +PL | 93.57±2.24 | 53.82±2.51 | 50.58±9.50 | 53.91±2.78 | 62.97 |
| +PLClf | 94.67±1.38 | 63.64±1.31 | 67.90±2.21 | 73.34±1.00 | 74.89 |
| +T3A | 97.52±1.99 | 65.32±2.24 | 70.70±3.48 | 75.51±1.75 | 77.26 |
| +TAST | 99.17±0.60 | 65.87±1.90 | 68.13±1.76 | 75.92±1.75 | 77.27 |
| +TAST-BN | 92.60±8.66 | 64.75±1.29 | 67.27±3.14 | 76.23±3.73 | 75.21 |
| Base | 95.76 ± 0.00 | 66.31 ± 0.00 | 70.07 ± 0.00 | 74.64 ± 0.00 | 76.7 |
| +UniDG | **99.79 ± 0.09** ↑ 4.03 | **68.78 ± 0.24** ↑ 2.47 | **75.01 ± 0.30** ↑ 4.94 | **79.93 ± 0.24** ↑ 5.29 | **80.88 ± 0.14** ↑ 4.18 |

Table 8: Full results using classifiers trained by ERM for Table 2 on PACS. We use ResNet-18 as a backbone network.

| Method | A | C | P | S | Avg |
|---|---|---|---|---|---|
| ERM | 77.78±0.81 | 75.09±1.22 | 95.19±0.29 | 69.11±1.22 | 79.29 |
| +Tent | 82.21±1.07 | 81.20±0.51 | 95.32±0.33 | 76.82±1.97 | 83.89 |
| +TentBN | 78.89±0.67 | 77.45±0.82 | 95.77±0.40 | 70.89±2.75 | 80.75 |
| +TentClf | 78.16±1.05 | 75.01±1.53 | 95.50±0.35 | 65.60±5.96 | 78.57 |
| +SHOT | 81.09±0.86 | 79.68±0.91 | 96.18±0.27 | 72.48±2.04 | 82.36 |
| +SHOTIM | 81.10±0.90 | 79.66±0.95 | 96.18±0.27 | 72.35±2.03 | 82.33 |
| +PL | 76.42±4.89 | 61.05±5.48 | 95.70±0.56 | 50.75±8.79 | 70.98 |
| +PLClf | 79.09±1.41 | 75.46±2.93 | 95.43±0.32 | 62.48±7.31 | 78.11 |
| +T3A | 78.81±0.97 | 77.14±1.20 | 95.92±0.36 | 71.44±1.63 | 80.83 |
| +TAST | 80.56±0.53 | 78.26±0.99 | 96.44±0.20 | 72.52±0.77 | 81.94 |
| +TAST-BN | 86.49±2.20 | 83.70±2.57 | 97.23±0.11 | 80.85±1.42 | 87.07 |
| Base | 76.51 ± 0.00 | 73.72 ± 0.00 | 92.37 ± 0.00 | 76.08 ± 0.00 | 79.7 |
| +UniDG | **80.19 ± 0.13** ↑ 3.68 | **76.47 ± 0.37** ↑ 2.75 | **95.41 ± 0.15** ↑ 3.04 | **74.86 ± 0.38** ↑ -1.22 | **81.73 ± 0.06** ↑ 2.03 |

Table 9: Full results using classifiers trained by ERM for Table 2 on OfficeHome. We use ResNet-18 as a backbone network.

| Method | A | C | P | R | Avg |
|---|---|---|---|---|---|
| ERM | 55.19±0.49 | 47.76±1.02 | 72.22±0.53 | 73.21±0.89 | 62.10 |
| +Tent | 53.39±0.61 | 48.28±0.88 | 70.50±0.68 | 71.29±0.72 | 60.86 |
| +TentBN | 55.53±0.43 | 49.53±0.95 | 72.47±0.27 | 73.01±1.23 | 62.64 |
| +TentClf | 55.17±0.67 | 36.73±1.94 | 72.21±0.52 | 73.22±0.97 | 59.33 |
| +SHOT | 55.14±0.57 | 50.27±1.18 | 71.69±0.45 | 73.21±0.91 | 62.58 |
| +SHOTIM | 55.08±0.56 | 50.29±1.17 | 71.71±0.40 | 73.21±0.90 | 62.57 |
| +PL | 54.49±1.06 | 34.66±13.13 | 71.45±0.37 | 72.20±0.65 | 58.20 |
| +PLClf | 55.14±0.70 | 47.70±1.25 | 72.21±0.54 | 72.62±0.96 | 61.92 |
| +T3A | 55.10±0.74 | 49.56±1.14 | 74.10±0.55 | 74.07±1.18 | 63.21 |
| +TAST | 56.15±0.68 | 50.04±1.31 | 74.33±0.28 | 74.28±1.23 | 63.70 |
| +TAST-BN | 55.11±0.58 | 51.35±0.85 | 72.58±0.80 | 72.13±0.78 | 62.79 |
| Base | 46.91 ± 0.00 | 45.42 ± 0.00 | 65.06 ± 0.00 | 66.49 ± 0.00 | 56.0 |
| +UniDG | **49.02 ± 0.11** ↑ 2.11 | **47.33 ± 0.24** ↑ 1.91 | **69.14 ± 0.15** ↑ 4.08 | **68.17 ± 0.22** ↑ 1.68 | **58.42 ± 0.06** ↑ 2.42 |

Table 10: Full results using classifiers trained by ERM for Table 2 on TerraIncognita. We use ResNet-18 as a backbone network.

| Method | L100 | L38 | L43 | L46 | Avg |
|---|---|---|---|---|---|
| ERM | 37.18±2.46 | 36.12±4.20 | 53.18±1.27 | 36.02±1.37 | 40.62 |
| +Tent | 38.29±0.48 | 25.82±3.91 | 41.53±1.59 | 29.15±1.83 | 33.70 |
| +TentBN | 40.55±1.46 | 37.44±2.22 | 46.33±1.32 | 35.30±1.26 | 39.91 |
| +TentClf | 34.44±13.31 | 34.19±5.76 | 52.71±2.03 | 31.86±2.26 | 38.30 |
| +SHOT | 33.87±0.66 | 28.58±2.10 | 40.99±2.07 | 30.83±1.26 | 33.57 |
| +SHOTIM | 33.83±1.29 | 28.13±2.30 | 40.81±2.18 | 30.64±1.46 | 33.35 |
| +PL | 51.92±1.19 | 35.61±20.74 | 39.97±10.98 | 22.26±8.21 | 37.44 |
| +PLClf | 45.22±2.45 | 36.03±5.81 | 52.76±1.54 | 33.10±2.27 | 41.78 |
| +T3A | 36.22±1.89 | 40.08±1.98 | 50.72±1.02 | 33.79±1.25 | 40.20 |
| +TAST | 43.67±2.83 | 39.24±3.79 | 52.64±3.02 | 35.01±1.09 | 42.64 |
| +TAST-BN | 51.06±7.31 | 32.74±7.54 | 41.70±2.86 | 32.21±3.05 | 39.43 |
| Base | 45.03 ± 0.00 | 32.67 ± 0.00 | 47.54 ± 0.00 | 35.80 ± 0.00 | 40.3 |
| +UniDG | **54.34 ± 0.21** ↑ 9.31 | **51.48 ± 2.16** ↑ 18.81 | **48.27 ± 0.46** ↑ 0.73 | **37.64 ± 0.21** ↑ 1.84 | **47.93 ± 0.65** ↑ 7.63 |

Table 11: Full results using classifiers trained by ERM for Table 2 on VLCS. We use ResNet-50 as a backbone network.

| Method | C | L | S | V | Avg |
|---|---|---|---|---|---|
| ERM | 97.66±0.64 | 63.87±1.71 | 71.21±1.52 | 74.09±2.06 | 76.71 |
| +Tent | 92.36±2.44 | 58.46±3.29 | 67.84±2.03 | 73.19±2.68 | 72.96 |
| +TentBN | 85.36±3.49 | 58.35±3.46 | 66.47±2.71 | 68.42±2.11 | 69.65 |
| +TentClf | 97.61±0.58 | 63.67±2.10 | 68.77±1.27 | 73.16±1.31 | 75.80 |
| +SHOT | 98.72±1.50 | 46.82±2.57 | 55.70±1.78 | 67.04±2.88 | 67.07 |
| +SHOTIM | 98.65±1.46 | 46.54±2.32 | 55.81±2.32 | 66.73±2.82 | 66.93 |
| +PL | 98.48±0.34 | 53.45±2.82 | 59.45±9.24 | 66.24±8.63 | 69.41 |
| +PLClf | 97.63±0.64 | 63.36±2.10 | 69.74±0.78 | 71.86±4.53 | 75.65 |
| +T3A | 99.17±0.38 | 64.78±1.61 | 73.01±3.24 | 72.20±2.84 | 77.29 |
| +TAST | 99.35±0.30 | 65.64±1.78 | 73.63±3.58 | 72.01±2.68 | 77.66 |
| +TAST-BN | 96.09±2.40 | 60.22±6.08 | 65.78±6.51 | 71.99±5.90 | 73.52 |
| Base | 97.70 ± 0.00 | 63.20 ± 0.00 | 70.18 ± 0.00 | 78.82 ± 0.00 | 77.50 |
| +UniDG | **99.71 ± 0.05** ↑ 2.01 | **71.11 ± 0.02** ↑ 7.91 | **73.60 ± 0.38** ↑ 3.42 | **81.99 ± 0.06** ↑ 3.17 | **81.60 ± 0.08** ↑ 4.10 |

Table 12: Full results using classifiers trained by ERM for Table 2 on PACS. We use ResNet-50 as a backbone network.

| Method | A | C | P | S | Avg |
|---|---|---|---|---|---|
| ERM | 82.92±1.65 | 78.05±3.36 | 96.50±0.32 | 75.38±3.31 | 83.21 |
| +Tent | 82.54±1.32 | 84.90±1.35 | 95.45±0.93 | 77.74±1.36 | 85.16 |
| +TentBN | 82.75±2.01 | 79.50±2.26 | 96.78±0.20 | 75.73±3.22 | 83.69 |
| +TentClf | 83.00±1.87 | 77.86±4.20 | 96.55±0.36 | 73.25±6.14 | 82.66 |
| +SHOT | 84.67±1.70 | 80.17±1.39 | 96.58±0.52 | 74.86±2.95 | 84.07 |
| +SHOTIM | 84.62±1.79 | 80.24±1.41 | 96.54±0.46 | 75.16±2.88 | 84.14 |
| +PL | 84.59±5.51 | 76.35±2.57 | 96.41±0.68 | 69.54±11.22 | 81.72 |
| +PLClf | 83.88±2.00 | 78.93±3.68 | 96.53±0.40 | 73.96±6.08 | 83.33 |
| +T3A | 83.56±2.03 | 79.75±3.14 | 96.99±0.24 | 75.36±3.57 | 83.92 |
| +TAST | 83.85±2.05 | 79.15±3.03 | 96.93±0.27 | 76.49±3.13 | 84.11 |
| +TAST-BN | 87.11±2.04 | 88.50±1.93 | 97.79±0.47 | 83.23±1.42 | 89.16 |
| Base | 86.52 ± 0.00 | 71.75 ± 0.00 | 97.16 ± 0.00 | 75.57 ± 0.00 | 82.7 |
| +UniDG | **91.26 ± 0.11** ↑ 4.74 | **84.86 ± 0.63** ↑ 13.11 | **98.13 ± 0.05** ↑ 0.97 | **81.77 ± 0.56** ↑ 6.2 | **89.00 ± 0.30** ↑ 6.30 |

Table 13: Full results using classifiers trained by ERM for Table 2 on OfficeHome. We use ResNet-50 as a backbone network.

| Method | A | C | P | R | Avg |
|---|---|---|---|---|---|
| ERM | 61.32±0.69 | 53.44±1.11 | 75.84±1.10 | 77.90±0.92 | 67.13 |
| +Tent | 60.98±0.67 | 53.94±1.24 | 74.49±0.71 | 75.75±0.53 | 66.29 |
| +TentBN | 62.63±0.45 | 54.90±1.17 | 76.20±1.09 | 77.92±1.01 | 67.91 |
| +TentClf | 61.35±0.73 | 52.72±1.40 | 75.23±1.05 | 77.86±1.07 | 66.79 |
| +SHOT | 61.91±0.33 | 55.58±0.91 | 75.49±1.54 | 77.60±0.80 | 67.65 |
| +SHOTIM | 61.84±0.32 | 55.63±0.92 | 75.56±1.60 | 77.57±0.79 | 67.65 |
| +PL | 59.42±1.55 | 42.40±12.31 | 73.80±2.26 | 75.77±1.50 | 62.85 |
| +PLClf | 61.35±0.40 | 52.87±1.96 | 75.86±1.09 | 77.94±1.10 | 67.01 |
| +T3A | 61.91±0.59 | 55.07±1.14 | 77.39±1.38 | 78.67±0.61 | 68.26 |
| +TAST | 62.43±0.80 | 55.81±1.26 | 77.46±1.07 | 78.83±0.93 | 68.63 |
| +TAST-BN | 63.22±0.85 | 58.20±0.98 | 77.14±1.10 | 76.94±0.39 | 68.88 |
| Base | 57.67 ± 0.00 | 53.69 ± 0.00 | 73.90 ± 0.00 | 75.70 ± 0.00 | 65.2 |
| +UniDG | **62.84 ± 0.31** ↑5.17 | **55.86 ± 0.12** ↑2.17 | **78.16 ± 0.05** ↑4.26 | **78.69 ± 0.02** ↑2.99 | **68.89 ± 0.07** ↑3.69 |

Table 14: Full results using classifiers trained by ERM for Table 2 on TerraIncognita. We use ResNet-50 as a backbone network.

| Method | L100 | L38 | L43 | L46 | Avg |
|---|---|---|---|---|---|
| ERM | 46.84±1.96 | 43.24±2.51 | 53.32±1.92 | 40.30±1.93 | 45.93 |
| +Tent | 41.20±2.71 | 29.72±3.59 | 41.35±2.92 | 36.03±2.85 | 37.08 |
| +TentBN | 46.64±1.17 | 41.11±3.16 | 49.31±1.05 | 38.52±2.04 | 43.89 |
| +TentClf | 49.87±3.80 | 43.31±3.19 | 53.01±2.31 | 28.40±6.19 | 43.64 |
| +SHOT | 36.17±2.70 | 29.80±2.92 | 41.00±0.30 | 33.83±1.86 | 35.20 |
| +SHOTIM | 35.56±2.76 | 27.49±4.01 | 40.77±0.45 | 33.67±1.84 | 34.37 |
| +PL | 56.75±5.78 | 46.12±1.03 | 29.44±10.14 | 20.06±4.65 | 38.09 |
| +PLClf | 52.28±3.95 | 43.76±2.96 | 52.78±2.15 | 37.81±2.49 | 46.66 |
| +T3A | 45.13±1.26 | 44.67±2.56 | 52.52±0.78 | 40.13±2.31 | 45.61 |
| +TAST | 53.01±3.95 | 43.27±3.21 | 53.79±2.72 | 39.66±3.65 | 47.43 |
| +TAST-BN | 55.75±2.37 | 33.92±9.86 | 43.87±4.70 | 32.33±4.40 | 41.47 |
| Base | 49.70 ± 0.00 | 40.51 ± 0.00 | 56.45 ± 0.00 | 33.63 ± 0.00 | 45.1 |
| +UniDG | **64.37 ± 0.11** ↑14.67 | **46.87 ± 0.32** ↑6.36 | **57.37 ± 0.22** ↑0.92 | **42.82 ± 0.57** ↑9.19 | **52.86 ± 0.17** ↑7.76 |

Table 15: Full results using classifiers trained by ERM with different visual backbones for Table 3 on VLCS dataset.

| Method | C | L | S | V | Avg |
|---|---|---|---|---|---|
| ResNet-101 He et al. (2016) | 97.00 ± 0.00 | 65.22 ± 0.00 | 70.64 ± 0.00 | 73.60 ± 0.00 | 76.6 |
| +UniDG | **99.35 ± 0.05** ↑2.35 | **69.71 ± 0.30** ↑4.49 | **75.01 ± 0.13** ↑4.37 | **78.05 ± 0.30** ↑4.45 | **80.53 ± 0.16** ↑3.93 |
| ViT-B16 Dosovitskiy et al. (2020) | 98.50 ± 0.00 | 63.44 ± 0.00 | 74.79 ± 0.00 | 77.67 ± 0.00 | 78.6 |
| +UniDG | **99.94 ± 0.02** ↑1.44 | **70.07 ± 0.17** ↑6.63 | **82.20 ± 0.63** ↑7.41 | **82.07 ± 0.10** ↑4.4 | **83.57 ± 0.13** ↑4.97 |
| DeiT Touvron et al. (2021) | 97.08 ± 0.00 | 66.96 ± 0.00 | 74.64 ± 0.00 | 79.27 ± 0.00 | 79.5 |
| +UniDG | **99.94 ± 0.02** ↑2.86 | **69.49 ± 0.19** ↑2.53 | **83.69 ± 0.13** ↑9.05 | **87.39 ± 0.28** ↑8.12 | **85.13 ± 0.05** ↑5.63 |
| HViT Dosovitskiy et al. (2020) | 96.73 ± 0.00 | 64.38 ± 0.00 | 75.40 ± 0.00 | 79.90 ± 0.00 | 79.1 |
| +UniDG | **99.53 ± 0.09** ↑2.8 | **69.49 ± 0.05** ↑5.11 | **79.27 ± 0.16** ↑3.87 | **85.82 ± 0.22** ↑5.92 | **83.53 ± 0.12** ↑4.43 |
| Swin Transformer Liu et al. (2021b) | 94.79 ± 0.00 | 65.27 ± 0.00 | 79.82 ± 0.00 | 80.04 ± 0.00 | 80.0 |
| +UniDG | **100.00 ± 0.00** ↑5.21 | **69.51 ± 0.35** ↑4.24 | **83.79 ± 0.32** ↑3.97 | **86.81 ± 0.15** ↑6.77 | **85.03 ± 0.10** ↑5.03 |
| MobileNet V3 Howard et al. (2019) | 72.8 ± 0.0 | 57.6 ± 0.0 | 58.3 ± 0.0 | 74.4 ± 0.0 | 65.8 |
| +UniDG | **93.8 ± 0.3** ↑21.0 | **63.2 ± 0.1** ↑5.6 | **69.1 ± 0.2** ↑10.8 | **78.8 ± 0.1** ↑4.4 | **76.2 ± 0.1** ↑10.4 |
| ConvNeXt Liu et al. (2022) | 97.97 ± 0.00 | 68.09 ± 0.00 | 78.37 ± 0.00 | 73.64 ± 0.00 | 79.5 |
| +UniDG | **100.00 ± 0.00** ↑2.03 | **72.97 ± 0.69** ↑4.88 | **84.77 ± 0.09** ↑6.4 | **85.50 ± 0.47** ↑11.86 | **85.81 ± 0.25** ↑6.31 |
| ViT-L16 Dosovitskiy et al. (2020) | 97.88 ± 0.00 | 61.74 ± 0.00 | 71.90 ± 0.00 | 73.94 ± 0.00 | 76.4 |
| +UniDG | **100.00 ± 0.00** ↑2.12 | **69.05 ± 0.38** ↑7.31 | **77.89 ± 0.55** ↑5.99 | **85.71 ± 0.44** ↑11.77 | **83.16 ± 0.23** ↑6.76 |
| Mixer-B16 Tolstikhin et al. (2021) | 93.46 ± 0.00 | 58.87 ± 0.00 | 70.26 ± 0.00 | 73.23 ± 0.00 | 74.0 |
| +UniDG | **99.20 ± 0.08** ↑5.74 | **67.69 ± 0.47** ↑8.82 | **79.14 ± 0.74** ↑8.88 | **79.13 ± 0.18** ↑5.9 | **81.29 ± 0.23** ↑7.29 |
| Mixer-L16 Tolstikhin et al. (2021) | 97.61 ± 0.00 | 61.65 ± 0.00 | 75.55 ± 0.00 | 77.42 ± 0.00 | 78.1 |
| +UniDG | **99.91 ± 0.04** ↑2.3 | **65.80 ± 0.28** ↑4.15 | **82.53 ± 0.06** ↑6.98 | **83.94 ± 0.05** ↑6.52 | **83.05 ± 0.09** ↑4.95 |

Table 16: Full results using classifiers trained by ERM with different visual backbones for Table 3 on PACS dataset.

| Method | A | C | P | S | Avg |
|---|---|---|---|---|---|
| ResNet-101 He et al. (2016) | 83.83 ± 0.00 | 82.30 ± 0.00 | 97.98 ± 0.00 | 79.29 ± 0.00 | 85.9 |
| +UniDG | **87.29 ± 0.14** ↑ 3.46 | **85.61 ± 0.09** ↑ 3.31 | **98.58 ± 0.00** ↑ 0.6 | **81.91 ± 0.43** ↑ 2.62 | **88.35 ± 0.06** ↑ 2.45 |
| ViT-B16 Dosovitskiy et al. (2020) | 85.54 ± 0.00 | 82.62 ± 0.00 | 98.88 ± 0.00 | 54.20 ± 0.00 | 80.3 |
| +UniDG | **89.85 ± 0.20** ↑ 4.31 | **87.28 ± 0.15** ↑ 4.66 | **99.78 ± 0.04** ↑ 0.9 | **64.84 ± 2.01** ↑ 10.64 | **85.44 ± 0.54** ↑ 5.14 |
| DeiT Touvron et al. (2021) | 90.73 ± 0.00 | 83.74 ± 0.00 | 99.03 ± 0.00 | 77.99 ± 0.00 | 87.9 |
| +UniDG | **94.55 ± 0.18** ↑ 3.82 | **90.60 ± 0.51** ↑ 6.86 | **99.70 ± 0.00** ↑ 0.67 | **85.41 ± 0.55** ↑ 7.42 | **92.57 ± 0.31** ↑ 4.67 |
| HViT Dosovitskiy et al. (2020) | 93.53 ± 0.00 | 81.66 ± 0.00 | 98.65 ± 0.00 | 85.56 ± 0.00 | 89.9 |
| +UniDG | **96.24 ± 0.12** ↑ 2.71 | **88.84 ± 0.33** ↑ 7.18 | **99.63 ± 0.04** ↑ 0.98 | **89.26 ± 0.08** ↑ 3.7 | **93.49 ± 0.10** ↑ 3.59 |
| Swin Transformer Liu et al. (2021b) | 93.11 ± 0.00 | 85.82 ± 0.00 | 99.48 ± 0.00 | 83.97 ± 0.00 | 90.6 |
| +UniDG | **97.38 ± 0.03** ↑ 4.27 | **90.65 ± 0.49** ↑ 4.83 | **99.78 ± 0.04** ↑ 0.3 | **89.50 ± 0.30** ↑ 5.53 | **94.33 ± 0.16** ↑ 3.73 |
| MobileNet V3 Howard et al. (2019) | 78.8 ± 0.0 | 77.7 ± 0.0 | 94.2 ± 0.0 | 66.5 ± 0.0 | 79.3 |
| +UniDG | **84.9 ± 0.2** ↑ 6.1 | **83.5 ± 0.2** ↑ 5.8 | **97.7 ± 0.1** ↑ 3.5 | **74.8 ± 0.3** ↑ 8.3 | **85.3 ± 0.4** ↑ 6.0 |
| ConvNeXt Liu et al. (2022) | 95.73 ± 0.00 | 85.34 ± 0.00 | 99.25 ± 0.00 | 90.81 ± 0.00 | 92.8 |
| +UniDG | **98.21 ± 0.09** ↑ 2.48 | **90.23 ± 0.50** ↑ 4.89 | **99.93 ± 0.00** ↑ 0.68 | **92.88 ± 0.16** ↑ 2.07 | **95.31 ± 0.16** ↑ 2.51 |
| ViT-L16 Dosovitskiy et al. (2020) | 93.17 ± 0.00 | 88.06 ± 0.00 | 99.55 ± 0.00 | 83.30 ± 0.00 | 91.0 |
| +UniDG | **98.19 ± 0.02** ↑ 5.02 | **92.54 ± 0.11** ↑ 4.48 | **99.98 ± 0.02** ↑ 0.43 | **90.21 ± 0.36** ↑ 6.91 | **95.23 ± 0.12** ↑ 4.23 |
| Mixer-B16 Tolstikhin et al. (2021) | 76.02 ± 0.00 | 71.59 ± 0.00 | 94.39 ± 0.00 | 62.09 ± 0.00 | 76.0 |
| +UniDG | **84.16 ± 0.35** ↑ 8.14 | **82.16 ± 0.38** ↑ 10.57 | **96.96 ± 0.05** ↑ 2.57 | **65.91 ± 0.10** ↑ 3.82 | **82.30 ± 0.13** ↑ 6.3 |
| Mixer-L16 Tolstikhin et al. (2021) | 84.93 ± 0.00 | 82.89 ± 0.00 | 98.73 ± 0.00 | 73.66 ± 0.00 | 85.1 |
| +UniDG | **91.74 ± 0.16** ↑ 6.81 | **85.89 ± 0.38** ↑ 3.0 | **99.38 ± 0.07** ↑ 0.65 | **76.86 ± 0.33** ↑ 3.2 | **88.47 ± 0.16** ↑ 3.37 |

Table 17: Full results using classifiers trained by ERM with different visual backbones for Table 3 on OfficeHome dataset.

| Method | A | C | P | R | Avg |
|---|---|---|---|---|---|
| ResNet-101 He et al. (2016) | 61.23 ± 0.00 | 55.18 ± 0.00 | 75.56 ± 0.00 | 77.74 ± 0.00 | 67.4 |
| +UniDG | **65.00 ± 0.28** ↑ 3.77 | **58.17 ± 0.51** ↑ 2.99 | **78.60 ± 0.12** ↑ 3.04 | **79.54 ± 0.02** ↑ 1.8 | **70.33 ± 0.20** ↑ 2.93 |
| ViT-B16 Dosovitskiy et al. (2020) | 71.37 ± 0.00 | 60.17 ± 0.00 | 84.40 ± 0.00 | 86.60 ± 0.00 | 75.6 |
| +UniDG | **78.29 ± 0.32** ↑ 6.92 | **66.46 ± 0.39** ↑ 6.29 | **89.10 ± 0.16** ↑ 4.7 | **89.99 ± 0.12** ↑ 3.39 | **80.96 ± 0.05** ↑ 5.36 |
| DeiT Touvron et al. (2021) | 75.28 ± 0.00 | 62.69 ± 0.00 | 84.23 ± 0.00 | 85.86 ± 0.00 | 77.0 |
| +UniDG | **78.30 ± 0.07** ↑ 3.02 | **66.56 ± 0.46** ↑ 3.87 | **85.82 ± 0.07** ↑ 1.59 | **87.14 ± 0.06** ↑ 1.28 | **79.46 ± 0.10** ↑ 2.46 |
| HViT Dosovitskiy et al. (2020) | 74.67 ± 0.00 | 70.85 ± 0.00 | 86.37 ± 0.00 | 87.44 ± 0.00 | 79.8 |
| +UniDG | **76.67 ± 0.11** ↑ 2.0 | **72.27 ± 0.54** ↑ 1.42 | **87.33 ± 0.14** ↑ 0.96 | **89.08 ± 0.22** ↑ 1.64 | **81.34 ± 0.06** ↑ 1.54 |
| Swin Transformer Liu et al. (2021b) | 80.74 ± 0.00 | 72.22 ± 0.00 | 88.20 ± 0.00 | 89.30 ± 0.00 | 82.6 |
| +UniDG | **83.26 ± 0.17** ↑ 2.52 | **74.42 ± 0.03** ↑ 2.2 | **89.90 ± 0.10** ↑ 1.7 | **90.69 ± 0.05** ↑ 1.39 | **84.57 ± 0.06** ↑ 1.97 |
| MobileNet V3 Howard et al. (2019) | 56.7 ± 0.0 | 46.2 ± 0.0 | 68.9 ± 0.0 | 73.0 ± 0.0 | 61.2 |
| +UniDG | **61.2 ± 0.5** ↑ 4.5 | **50.7 ± 0.5** ↑ 4.5 | **73.1 ± 0.1** ↑ 4.2 | **75.4 ± 0.1** ↑ 2.4 | **65.1 ± 0.2** ↑ 3.9 |
| ConvNeXt Liu et al. (2022) | 83.88 ± 0.00 | 78.12 ± 0.00 | 90.62 ± 0.00 | 91.37 ± 0.00 | 86.0 |
| +UniDG | **87.68 ± 0.11** ↑ 3.8 | **80.95 ± 0.07** ↑ 2.83 | **92.25 ± 0.04** ↑ 1.63 | **93.23 ± 0.09** ↑ 1.86 | **88.53 ± 0.04** ↑ 2.53 |
| ViT-L16 Dosovitskiy et al. (2020) | 81.31 ± 0.00 | 71.56 ± 0.00 | 89.33 ± 0.00 | 90.71 ± 0.00 | 83.2 |
| +UniDG | **86.25 ± 0.29** ↑ 4.94 | **79.71 ± 0.20** ↑ 8.15 | **91.35 ± 0.16** ↑ 2.02 | **92.85 ± 0.11** ↑ 2.14 | **87.54 ± 0.17** ↑ 4.34 |
| Mixer-B16 Tolstikhin et al. (2021) | 30.69 ± 0.00 | 47.97 ± 0.00 | 70.33 ± 0.00 | 60.90 ± 0.00 | 52.5 |
| +UniDG | **32.60 ± 0.24** ↑ 1.91 | **53.68 ± 0.73** ↑ 5.71 | **77.18 ± 0.13** ↑ 6.85 | **67.26 ± 0.22** ↑ 6.36 | **57.68 ± 0.27** ↑ 5.18 |
| Mixer-L16 Tolstikhin et al. (2021) | 67.61 ± 0.00 | 62.54 ± 0.00 | 82.49 ± 0.00 | 68.10 ± 0.00 | 70.2 |
| +UniDG | **71.78 ± 0.36** ↑ 4.17 | **68.41 ± 0.23** ↑ 5.87 | **87.08 ± 0.15** ↑ 4.59 | **75.04 ± 0.16** ↑ 6.94 | **75.58 ± 0.08** ↑ 5.38 |

Table 18: Full results using classifiers trained by ERM with different visual backbones for Table 3 on TerraIncognita dataset.

| Method | L100 | L38 | L43 | L46 | Avg |
|---|---|---|---|---|---|
| ResNet-101 He et al. (2016) | $37.44 \pm 0.00$ | $36.78 \pm 0.00$ | $57.08 \pm 0.00$ | $39.11 \pm 0.00$ | 42.6 |
| +UniDG | $\mathbf{50.65 \pm 0.47}$ ↑13.21 | $\mathbf{46.71 \pm 0.76}$ ↑9.93 | $\mathbf{59.87 \pm 0.58}$ ↑2.79 | $\mathbf{42.89 \pm 0.85}$ ↑3.78 | $\mathbf{50.03 \pm 0.46}$ ↑7.43 |
| ViT-B16 Dosovitskiy et al. (2020) | $53.97 \pm 0.00$ | $36.12 \pm 0.00$ | $48.46 \pm 0.00$ | $35.25 \pm 0.00$ | 43.4 |
| +UniDG | $\mathbf{65.50 \pm 1.36}$ ↑11.53 | $\mathbf{48.83 \pm 1.34}$ ↑12.71 | $\mathbf{51.61 \pm 0.27}$ ↑3.15 | $\mathbf{39.68 \pm 0.16}$ ↑4.43 | $\mathbf{51.40 \pm 0.22}$ ↑8.0 |
| DeiT Touvron et al. (2021) | $56.45 \pm 0.00$ | $42.20 \pm 0.00$ | $55.07 \pm 0.00$ | $43.40 \pm 0.00$ | 49.3 |
| +UniDG | $\mathbf{64.56 \pm 0.60}$ ↑8.11 | $\mathbf{51.00 \pm 1.93}$ ↑8.8 | $\mathbf{56.55 \pm 0.52}$ ↑1.48 | $\mathbf{44.32 \pm 0.13}$ ↑0.92 | $\mathbf{54.11 \pm 0.42}$ ↑4.81 |
| HViT Dosovitskiy et al. (2020) | $61.59 \pm 0.00$ | $46.13 \pm 0.00$ | $61.56 \pm 0.00$ | $42.11 \pm 0.00$ | 52.8 |
| +UniDG | $\mathbf{73.64 \pm 1.23}$ ↑12.05 | $\mathbf{54.52 \pm 0.60}$ ↑8.39 | $\mathbf{63.57 \pm 0.49}$ ↑2.01 | $\mathbf{48.57 \pm 0.98}$ ↑6.46 | $\mathbf{60.08 \pm 0.36}$ ↑7.28 |
| Swin Transformer Liu et al. (2021b) | $80.74 \pm 0.00$ | $72.22 \pm 0.00$ | $88.20 \pm 0.00$ | $89.30 \pm 0.00$ | 82.6 |
| +UniDG | $\mathbf{83.26 \pm 0.17}$ ↑2.52 | $\mathbf{74.42 \pm 0.03}$ ↑2.2 | $\mathbf{89.90 \pm 0.10}$ ↑1.7 | $\mathbf{90.69 \pm 0.05}$ ↑1.39 | $\mathbf{84.57 \pm 0.06}$ ↑1.97 |
| MobileNet V3 Howard et al. (2019) | $30.6 \pm 0.0$ | $26.4 \pm 0.0$ | $31.9 \pm 0.0$ | $31.2 \pm 0.0$ | 30.0 |
| +UniDG | $\mathbf{39.2 \pm 0.3}$ ↑8.6 | $\mathbf{29.8 \pm 1.3}$ ↑3.4 | $\mathbf{31.5 \pm 0.1}$ ↑-0.4 | $\mathbf{38.3 \pm 0.8}$ ↑8.2 | $\mathbf{34.7 \pm 0.2}$ ↑4.7 |
| ConvNeXt Liu et al. (2022) | $67.70 \pm 0.00$ | $55.75 \pm 0.00$ | $67.51 \pm 0.00$ | $53.24 \pm 0.00$ | 61.0 |
| +UniDG | $\mathbf{77.05 \pm 0.18}$ ↑9.35 | $\mathbf{59.84 \pm 0.77}$ ↑4.09 | $\mathbf{68.25 \pm 0.20}$ ↑0.74 | $\mathbf{56.23 \pm 0.48}$ ↑2.99 | $\mathbf{65.34 \pm 0.27}$ ↑4.34 |
| ViT-L16 Dosovitskiy et al. (2020) | $48.85 \pm 0.00$ | $42.48 \pm 0.00$ | $51.35 \pm 0.00$ | $38.18 \pm 0.00$ | 45.2 |
| +UniDG | $\mathbf{64.57 \pm 1.42}$ ↑15.72 | $\mathbf{54.77 \pm 0.38}$ ↑12.29 | $\mathbf{55.06 \pm 0.11}$ ↑3.71 | $\mathbf{41.06 \pm 0.31}$ ↑2.88 | $\mathbf{53.86 \pm 0.42}$ ↑8.66 |
| Mixer-B16 Tolstikhin et al. (2021) | $26.84 \pm 0.00$ | $26.95 \pm 0.00$ | $30.07 \pm 0.00$ | $22.37 \pm 0.00$ | 26.6 |
| +UniDG | $\mathbf{46.89 \pm 2.11}$ ↑20.05 | $\mathbf{53.87 \pm 0.18}$ ↑26.92 | $\mathbf{37.97 \pm 0.99}$ ↑7.9 | $\mathbf{26.10 \pm 0.17}$ ↑3.73 | $\mathbf{41.21 \pm 0.48}$ ↑14.61 |
| Mixer-L16 Tolstikhin et al. (2021) | $38.18 \pm 0.00$ | $26.72 \pm 0.00$ | $47.23 \pm 0.00$ | $33.97 \pm 0.00$ | 36.5 |
| +UniDG | $\mathbf{57.20 \pm 0.75}$ ↑19.02 | $\mathbf{36.55 \pm 5.39}$ ↑9.83 | $\mathbf{49.70 \pm 0.23}$ ↑2.47 | $\mathbf{36.47 \pm 0.38}$ ↑2.5 | $\mathbf{44.98 \pm 1.40}$ ↑8.48 |

