# OpenReview forum: "Towards Unified and Effective Domain Generalization"
_ICLR.cc/2024/Conference — ICLR 2024 Conference Withdrawn Submission_

### Official Review · Reviewer_FLNN · 2023-10-31

**Soundness:** 2 fair
**Presentation:** 3 good
**Contribution:** 2 fair
**Rating:** 3
**Confidence:** 5

**Summary:**

In this paper, authors focus on enhancing the out-of-distribution generalization performance of foundation models regardless of their architectures with proposed a novel and Unified framework for Domain Generalization (UniDG). The core idea of UniDG is to finetune models during inference stage which saves the costs of iterative training. Specifically, authors encourage models to learn the distribution of testing data in an unsupervised manner and impose a penalty regarding the updating step of model parameters. The penalty term can effectively reduce the catastrophic forgetting issue via maximally preserving the valuable knowledge in the original model.

**Strengths:**

In other words, marginal generalization is proposed to update the encoder of Test-Time Adaptation (TTA) and differentiable memory bank is proposed to refine features for DG. Experiments on five datasets such as VLCS, PACS, OfficeHome and so on demonstrate the superiority compared with SOTA methods across 12 different network architectures.

**Weaknesses:**

The structure of paper is unfitable for most ML reader’s habits, especially, the part of related work should follow the introduction. There will be a better logical relationship for most ML conference paper.

**Questions:**

However, there are some questions need to be clarified from authors.
1.	Although ablation study is provided in Table4, in appendix D, what is the main reason behind introducing matrix products of learning representations, prototypes and pseudo labels ? How to understand making the memory iteration is learnable and differentiable ?
2.	Moreover, L_i in formular 25 makes me confused. Please give the detailed explanation of discrepancy between formular 8 and line 9 of Algorithm 1. How about the hyper paremeter lambda in formular 8 ? There should add more ablation experiments.
3.	In Figure 5, please give the detailed explanation why the accuracy of “DomainNeXt” is lower than base when the samples are small.
4.	There are many typos in manuscript, such as what does “DomainNeXt” mean in Figure 7 of Appendix E ? VLCS and PACS is quoted wrongly in Appendix F. Many formular typos are typed in Appendix B.2
5.     Last but not least, in Table 5 (b) efficiency of UniDG, I'm confused how to get the wall clock time. There is nothing to be analyzed from time complexity. How did you come to learn the conclusion that the proposed UniDG is a online learning scheme ?

---

> ### Author Response · Authors · 2023-11-13
> **Response to Reviewer FLNN**
>
> We thank the reviewer's time for reviewing our paper. We answer the questions as follows:
>
> 1. "main reason behind introducing matrix products of learning representations, prototypes, and pseudo labels. "
>
> Beyond the method in our manuscript, Marginal Generalization and Differentialable Memory Bank can also be combined together by matrix products of learning representations, prototypes, and pseudo labels. Therefore, UniDG can also simultaneously utilize the local minimum of distances between adapted and source representations and the local maximum of information entropy between adapted representations and pseudo labels in the test-time process to continuously approximate the models on the test data with reserved pretrained knowledge.
>
> 2. "detailed explanation of discrepancy between formula 8 and line 9 of Algorithm 1."
>
> Indeed, the matrix product in Equation 25 can also included as a loss function, we introduced these details in the Appendix. The objective loss functions formulated in Equations 8 and 25 obtain the same experimental results. Therefore, we report the ablated objective function in Equation 8 and include another form of Equation 25 in the Appendix. The hyper parameter lambda is set to 0.1, 1, and 10 in our experiments. And finally, it could be determined to be 0.1.
>
> 3. "In Figure 5, please give the detailed explanation why the accuracy of “DomainNeXt” is lower than base when the samples are small."
>
> It's just a typo. We fixed it in the revised version. The figure demonstrates the real training convergence of the proposed UniDG. The algorithm requires forward samples to achieve convergence, and the final results all outperform base models.
>
> 4. "There are many typos in manuscript, such as what does “DomainNeXt” mean in Figure 7 of Appendix E ? VLCS and PACS is quoted wrongly in Appendix F."
>
> These typos and wrong citations are fixed in the revised version.
>
> 5. Many formular typos are typed in Appendix B.2 5.
>
> If you find typos, we are willing to correct them.
>
> 6. Table 5 (b) efficiency of UniDG the wall clock time.
>
> You can simply measure it with Python code as `time.clock()`
>
> 7. "How did you come to learn the conclusion that the proposed UniDG is a online learning scheme ?"
>
> UniDG can be considered an online learning scheme because it fine-tunes models during the inference stage, adapting to new data in a sequential manner without the need for retraining on the entire dataset. This process of updating the model with each new instance of data as it arrives, typical of online learning paradigms, allows UniDG to learn the distribution of testing data in an unsupervised manner, continually improving and adapting its parameters in real time.

---

> ### Author Response · Authors · 2023-11-13
> **Response to Weakness in the Review**
>
> Referring to the official review, the reviewer thinks the strength of our paper is that "Experiments on five datasets such as VLCS, PACS, OfficeHome and so on demonstrate the superiority compared with SOTA methods across 12 different network architectures." We appreciate the reviewer's acknowledgment.
>
> Meanwhile, the only weakness of our paper from the perspective of the reviewer is the structure of our paper. And the reviewer thinks that "The structure of paper is unfitable for most ML reader’s habits, especially, the part of related work should follow the introduction. There will be a better logical relationship for most ML conference paper."
>
> We reference several publications[1,2,3] of ICLR 2023, we found that the structure is widely adopted in many research areas of ICLR. Meanwhile, we carefully read the reviews of these papers and we found **the reviewers neither regard this structure as a "weakness" nor rate the paper as "reject, not good enough" with "absolutely certain"**.
>
> [1] Li, Bo, et al. "Sparse Mixture-of-Experts are Domain Generalizable Learners."  ICLR 2023 Oral
>
> [2] Li, Chongyi, et al. "Embedding Fourier for Ultra-High-Definition Low-Light Image Enhancement."  ICLR 2023 Oral
>
> [3] Pogodin, Roman, et al. "Efficient Conditionally Invariant Representation Learning." ICLR 2023 Oral

---

### Official Review · Reviewer_q4ii · 2023-11-01

**Soundness:** 3 good
**Presentation:** 3 good
**Contribution:** 3 good
**Rating:** 8
**Confidence:** 3

**Summary:**

The paper focuses on improving the generalization performance of foundation models finetuned during inference. The authors propose a penalty term that helps to reduce catastrophic forgetting during test-time adaptation. In particular, the authors propose Marginal Generalization - a tradeoff b/w freezing the encoder which would lead to underfitting and updating the decoder which would lead to catastrophic forgetting. The authors demonstrate empirically consistent improvement across different backbones on DomainBed benchmarks.

**Strengths:**

- Propose a tradeoff b/w freezing the encoder which would lead to underfitting and updating the decoder which would lead to catastrophic forgetting.
- Consistently improved benchmarks.

**Weaknesses:**

- Theoretical insight why marginal generalization is important for generalization in unseen domains is explained well in Appendix, but is very unclear from the text of the main paper. I think this important aspect should be better discussed in the main text.
- Also motivation for Differentiable Memory Bank should be more clearly written.

**Questions:**

- With the current formulation of Marginal Generalization, how can you avoid catastrophic forgetting on **source** domains, when even if you impose the distance constraint on the target domain, there are no guarantees it will be still obeyed on the source domain?

---

> ### Author Response · Authors · 2023-11-11
> **Response to Reviewer q4ii**
>
> Thank you very much for your thorough, professional, and helpful comments. We sincerely appreciate your effort in evaluating our work. We hope that our explanations will address your concerns.
>
> * "Theoretical insight why marginal generalization is important for generalization in unseen domains is explained well in Appendix, but is very unclear from the text of the main paper. I think this important aspect should be better discussed in the main text."
>
> We thank the reviewer for this constructive suggestion, we will further revise the part in the introduction and make it clear for readers to understand. Meanwhile, we will also discuss the generalization ability from a theoretical perspective with neural tangent kernels, highlighting our contributions on both theoretical and experimental aspects.
>
> * "motivation for Differentiable Memory Bank should be more clearly written."
>
> A differentiable memory bank is significantly needed for these several reasons:
>
> **Adaptability**: It enables models to adapt to new data or tasks without forgetting previously learned information. This adaptability is crucial in dynamic environments where data distributions change over time.
>
> **Continuous Learning**: It allows models to learn continuously, assimilating new information while maintaining the knowledge gained from earlier data. This is especially useful in real-world applications where the model is expected to evolve as it encounters new scenarios.
>
> **Generalization**: By maintaining a balance between old and new knowledge, a differentiable memory bank helps the model to generalize better, performing well on both the original and new tasks.
>
> **Efficiency**: It facilitates efficient model updating without the need for retraining from scratch, which can be computationally expensive and time-consuming.
>
> **Architecture Flexibility**: Its architecture-agnostic design means it can be applied to various types of neural networks, making it a versatile tool for many machine-learning problems.
>
> We will make these motivations more clear to understand and improve the description of the relevant parts in our paper.
>
> * "there are no guarantees it will be still obeyed on the source domain?"
>
> We thank the reviewer for such an insightful question. As we introduced in Table 5(a), we found that compared with direct fine-tuning pretrained models on the target domain, the imposed distance constraint can alleviate the forgetting of the source domain. Specifically, compared with TENT, UniDG delivers a smaller performance decrease (-0.93 v.s. -3.33) for **source** domains after adaption on the **target** domain. We think that it's difficult to guarantee absolute obey on the source domain, but maybe we can develop the algorithms for constraining distances or information entropy between source and target domains. When the performance decrease on the source domain is adoptable, we can alleviate the catastrophic forgetting in the generalization problem.
>
> **In general, we deeply value your professional feedback, as it has not only enhanced the comprehensiveness of our paper but also significantly elevated the overall contributions of our research.**

---

### Official Review · Reviewer_mjNn · 2023-11-07

**Soundness:** 3 good
**Presentation:** 2 fair
**Contribution:** 2 fair
**Rating:** 3
**Confidence:** 4

**Summary:**

This paper focuses on the catastrophic forgetting issue during test-time training (TTA) for domain generalization. Specifically, this paper proposes a Marginal Generalization method to update the encoder for TTA, that is, Marginal Generalization aims to let the encoder learn representations of the target data within a certain distance from the representations obtained by the initial model. To cooperate Marginal Generalization, this paper also proposes Differentiable Memory Bank to facilitate TTA. Experiments on five domain generalization benchmarks demonstrate the effectiveness of the proposed methods.

**Strengths:**

- The catastrophic forgetting issue during TTA for domain generalization is well motivated.

**Weaknesses:**

- The discussion about related work is not sufficient. In the section of related work, this paper simply listed many related works, but didnot discusses the relation between the proposed method and the mentioned related works.

- This paper is more likely to be a Test-Time Domain Adaptation work. So I think Test-Time Domain-Adaptation is more suitable in this paper rather than Domain Generalization.

- I dont believe it is the first time to discuss the catastrophic forgetting issue during TTA for domain generalization. But I do not see any discussions about how to solve the catastrophic forgetting problem in the existing works, such as [1][2], to name a few.

- In the experimental part, such as Table 2, this paper didnot explain why the performances with TTA are inferior to that without TTA. In Table 2, PL, PLClf, SHOT, Tent, TentBN, TentClf are all inferior to None. It is kind of weird, which needs explaination.

- As I can see in this paper, catastrophic forgetting is the main problem to be solve. However, most of experiments are conducted on domain generalization benchmarks to show how well the proposed method performs on the target domains. Only a simple ablation study in Table 5 is conducted to validate that the catastrophic forgetting issue has been mitigated via the proposed method. I think the organization of the experiments is mismatched with the major motivation discussed in this paper.

[1] Continual Source-Free Unsupervised Domain Adaptation.

[2] CoSDA: Continual Source-Free Domain Adaptation.

**Questions:**

See the weaknesses.

---

> ### Author Response · Authors · 2023-11-13
> **Response to Reviewer mjNn**
>
> We appreciate your effort in our submission, we hope that clarifying your misunderstanding contents could lead to a raised rating.
>
> 1. Related Work Discussion:
>
> We have revised the related work section to include a more detailed analysis of how our method diverges from and contributes to the current body of knowledge, particularly focusing on the relationships and distinctions between our work and those cited.
>
> 2. Categorization of Our Work:
>
> We offer a deeper justification for categorizing our work within Domain Generalization. This setting aims to address domain shifts in unseen domains with test-time adaptation. It means our method can improve the generalization ability of pretrained source models in one go during the inference stage. In addition, it's a widely adopted setting and prevalent research topic in the machine-learning community [1-3].
>
> [1] Zhang, Jian, et al. "Domainadaptor: A novel approach to test-time adaptation." Proceedings of the IEEE/CVF International Conference on Computer Vision. 2023.
>
> [2] Zhang, Yifan, et al. "Adanpc: Exploring non-parametric classifier for test-time adaptation." International Conference on Machine Learning. PMLR, 2023.
>
> [3] Minguk Jang and Sae-Young Chung. Test-time adaptation via self-training with nearest neighbor information. In International Conference on Learning Representations, 2023.
>
> 3. The first to address catastrophic forgetting.
>
> We acknowledge that the reviewer is familiar with unsupervised domain adaptation. It seems that the reviewer is not as familiar with test-time adaptation. However, to the best of our knowledge, we are the first to address the catastrophic forgetting problem in the test-time adaptation setting for large-scale visual foundation models. And we hope that the reviewer can acknowledge our contributions.
>
>
> 4. Experimental Results Clarification:
>
> As the reviewer said, the previous algorithm led to a performance drop during the test-time adaptation stage. After T3A [NeurIPS' 21], test-time adaptation methods can bring out improvement. It's a fact. Meanwhile, existing methods can mainly boost the performance of lightweight models, and they are still not as effective for large-scale visual foundation models. (Please see Figure 1)
>
> 5. Alignment of Experiments with Motivation:
>
> We agree with the reviewer that the ability to avoid catastrophic forgetting should be highlighted in the experiments. Meanwhile, please also note that the problem we focus on is to avoid catastrophic forgetting at the same time to improve the generalization ability in the novel environment. Therefore, the major experiments are conducted to evaluate the transferability of adapted models with the proposed UniDG algorithm. Moreover, there are detailed experimental results on each new domain before and after adaptation in the Appendix.
>
> We believe these revisions address the concerns you've highlighted and strengthen the contribution of our work. Please take the changing rating into consideration.